# Any1 is a phospholipid scramblase involved in endosome biogenesis

Jieqiong Gao[1], Rico Franzkoch[2,4]*, Cristian Rocha-Roa[3]*, Olympia Ekaterini Psathaki[4], Michael Hensel[2,5], Stefano Vanni[3,6], and Christian Ungermann[1,5]

Endosomes are central organelles in the recycling and degradation of receptors and membrane proteins. Once endocytosed, such proteins are sorted at endosomes into intraluminal vesicles (ILVs). The resulting multivesicular bodies (MVBs) then fuse with the lysosomes, leading to the degradation of ILVs and recycling of the resulting monomers. However, the biogenesis of MVBs requires a constant lipid supply for efficient ILV formation. An ER–endosome membrane contact site has been suggested to play a critical role in MVB biogenesis. Here, we identify Any1 as a novel phospholipid scramblase, which functions with the lipid transfer protein Vps13 in MVB biogenesis. We uncover that Any1 cycles between the early endosomes and the Golgi and colocalizes with Vps13, possibly at a here-discovered potential contact site between lipid droplets (LDs) and endosomes. Strikingly, both Any1 and Vps13 are required for MVB formation, presumably to couple lipid flux with membrane homeostasis during ILV formation and endosome maturation.

## Introduction

In eukaryotic cells, the transport of proteins along the endomembrane system with its secretory and endolysosomal branches occurs in membrane-surrounded carriers such as vesicles and tubules. Such carriers form and fuse with organelles, where proteins are modified or sorted. These need to maintain their distinct identities and functions despite the constant membrane surface alterations (Bonifacino and Glick, 2004; Lee et al., 2004).

In the endolysosomal pathway, cell surface proteins such as receptors or nutrient transporters are internalized at the plasma membrane (PM) into endocytic vesicles and subsequently delivered to the early endosomes (EE). Proteins are then either sorted back to the PM via recycling endosomes or remain in the EE, which then matures into late endosomes (LE) (Huotari and Helenius, 2011; O'sullivan and Lindsay, 2020). This maturation process requires that the EEs continue to fuse with themselves or with endocytic vesicles to increase in size. EE and LE also fuse with Golgi-derived vesicles, which transport lysosomal hydrolases and other membrane proteins to the lysosome. During the maturation, the Retromer complex collects receptors such as the mannose-6-phosphate receptor into tubular carriers to allow for their reuse at the Golgi (Seaman, 2012). In addition, EE-to-LE conversion requires the exchange of the EE-specific Rab5 GTPase to the LE-specific Rab7 (Burd and Cullen, 2014; Borchers

et al., 2021). Maturation also results in a change of endosomal shape. To allow for the degradation of membrane proteins, the endosomal sorting complex required for transport (ESCRT) sorts these proteins into intraluminal vesicles (ILVs). Consequently, LEs round up into a spherical structure with multiple ILVs, now called multivesicular bodies (MVBs) (Hurley and Emr, 2006; Henne et al., 2011). These MVBs eventually fuse with the lysosome, where ILVs and their content are degraded by resident hydrolases into their precursors, which are exported from the lysosomal lumen to the cytosol for cellular reuse. However, several details regarding the lipid supply during MVB formation and the regulation of endosome maturation remain largely unknown. Membrane contact sites (MCSs), areas of close apposition between organelles, have been suggested to play essential roles in organelle biogenesis and growth by providing lipids via lipid transport proteins (Bieber et al., 2022; Chumpen Ramirez et al., 2023; Suzuki et al., 2024). Lipid droplets (LDs) are essential in the storage and distribution of lipids to control cellular metabolism and homeostasis (Olzmann and Carvalho, 2019; Walther et al., 2017; Zadoorian et al., 2023), and they form a distinct MCS with the ER. In mammalian cells, LDs also interact with tubules of the EE, which may be part of tripartite contacts with the ER, though the function of this MCS is currently unknown (Parton et al., 2020). A key protein proposed to control

[1]Department of Biology/Chemistry, Biochemistry Section, Osnabrück University, Osnabrück, Germany; [2]Department of Biology/Chemistry, Division of Microbiology, Osnabrück University, Osnabrück, Germany; [3]Department of Biology, University of Fribourg, Fribourg, Switzerland; [4]Integrated Bioimaging Facility, Center of Cellular Nanoanalytic Osnabrück (CellNanOs), Osnabrück University, Osnabrück, Germany; [5]Center of Cellular Nanoanalytic Osnabrück (CellNanOs), Osnabrück University, Osnabrück, Germany; [6]Swiss National Center for Competence in Research Bio-Inspired Materials, University of Fribourg, Fribourg, Switzerland.

*R. Franzkoch and C. Rocha-Roa contributed equally to this paper.   Correspondence to Christian Ungermann: cu@uos.de;   Jieqiong Gao: jieqiong.gao@uni-osnabrueck.de.

lipid supply at various MCSs is the bridge-like lipid transfer protein (BLTP), Vps13 (Dziurdzik and Conibear, 2021; Ugur et al., 2020; Melia and Reinisch, 2022). Recently, Vps13 was identified on yeast endosomes as a critical protein required for lipid transport at an ER–endosome MCS, thereby ensuring membrane supply for ESCRT-mediated sorting during MVB biogenesis (Suzuki et al., 2024). However, this protein-mediated lipid transport likely requires an endosomal scramblase to rebalance lipids between the two leaflets of the acceptor membrane (Ghanbarpour et al., 2021). This is the case for autophagosome biogenesis, where Atg2, a BLTP structurally similar to Vps13, facilitates phagophore expansion by delivering lipids from the ER at an ER–phagophore contact site. Importantly, the activity of Atg2 is coordinated with the lipid scramblase Atg9 on the phagophore (Ghanbarpour et al., 2021; Chumpen Ramirez et al., 2023; van Vliet et al., 2022; Matoba et al., 2020). Yet, which specific scramblase on the endosomes equilibrates lipids between the two leaflets of the endosomal membrane during endosome formation is still unknown.

Any1 (alias Cfs1) is a seven-helix membrane protein of the "PQ-loop family" and has a duplicated motif termed the "PQ-loop" (Jeźégou et al., 2012; Yamamoto et al., 2017). Yeast has five other PQ-loop family proteins (Ypq1, Ypq2, Ypq3, Ers1, and Ydr090c), which function as amino acid transporters at the vacuole (Jeźégou et al., 2012; Gao et al., 2005; Simpkins et al., 2016; Sekito et al., 2014; Manabe et al., 2016). Any1 has been suggested to localize to endosomes and trans-Golgi network membranes (Yamamoto et al., 2017). Interestingly, the any1Δ mutation suppressed growth defects and membrane trafficking defects of all phospholipid flippase mutants (Yamamoto et al., 2017). These flippases are P4-type ATPases, which generate membrane asymmetry at the Golgi and endosome and subsequently at the PM (Tanaka et al., 2011; Yamamoto et al., 2017). Any1 was therefore suggested to act as a regulator of flippase or scramblase at these organelles (Yamamoto et al., 2017). However, in contrast to other PQ-loop proteins, the function of Any1 and its closest human homolog, PQLC1 (SLC66A2), remained unclear.

Here, we identify Any1 as a phospholipid scramblase, which mainly localizes to the Golgi and partially to the EE, and cycling between these two organelles requires the Retromer complex. Deletion of Any1 reduces the number of MVBs and impairs the endocytic pathway. Consistently, mutants in Any1 within the lipid scrambling pathway abolish scramblase activity in silico and lead to an accumulation of EE and a reduced number of MVBs in vivo. Our data suggest that Any1 activity is coordinated with the function of the lipid transfer protein Vps13 at a potential LD/ER–endosomal MCS to rebalance lipids between endosomal membrane leaflets, thereby promoting endosomal membrane expansion.

## Results

### Any1 is required for the endocytic pathway

Given that Any1 has been suggested to localize to the endosomes and trans-Golgi network membranes (Yamamoto et al., 2017) (Fig. 1 A), we asked whether it functions in the biogenesis of the endolysosomal system. To address this, we first conducted growth assays to analyze the effects of the any1 deletion and tagging on cell growth. We therefore spotted cells in serial dilutions on plates containing 5 mM Zn²⁺, a stressor of vacuoles and thus endolysosomal transport (Fig. S1 A). We observed a clear growth defect in the any1Δ mutant, whereas cells expressing C-terminally tagged Any1 variants, such as mKate or mNeonGreen (mNeon), grew like wild-type (Fig. S1 A). These findings suggest that Any1 is important for endolysosomal transport and that the tagged versions of Any1 retain their functionality. We next determined the distribution of Any1 by analyzing the localization of mKate-tagged Any1 relative to functionally tagged markers of the Golgi or endosomes carrying a GFP or mNeon fluorophore. In particular, we analyzed Sec7 (trans-Golgi network), Vps21 (a Rab5-like protein at EE), and Ypt7 (a Rab7-like protein at LE) (Fig. 1 B). In agreement with previous findings (Yamamoto et al., 2017), we observed that Any1 strongly colocalized with Sec7 and less so with Vps21 (Fig. 1, B and C). However, Any1 did not colocalize with Ypt7 (Fig. 1, B and C), indicating that Any1 is present on the trans-Golgi network and EE, but not on the mature MVBs. To confirm whether Any1 is an endosomal protein, we analyzed the colocalization of mNeon-tagged Any1 with mCherry-tagged carboxypeptidase S (Cps1) in wild-type and vps4Δ cells. Cps1 is sorted at the MVB via ESCRTs into ILVs and accumulates in the vacuole lumen in wild-type cells (Gao et al., 2022). Loss of ESCRT proteins results in the accumulation of multilamellar structures, called Class E compartment, next to the vacuole, where all endosomal proteins and cargo proteins like Cps1 accumulate (Raymond et al., 1992; Rieder et al., 1996; Babst et al., 1998; Russell et al., 2012). Upon deletion of VPS4, Any1 partially colocalized with Cps1, which accumulated at Class E compartments (Fig. 1, D and E). Taken together, our data demonstrate that Any1 localizes to both the Golgi and endosomes.

The highly conserved Retromer complex mediates endosomes to Golgi retrieval of specific membrane proteins. These cargoes are concentrated in discrete regions of the endosomal membrane, where tubules form to transport cargo proteins back to the Golgi (Seaman et al., 1997, 1998). The yeast heteropentameric retromer complex can be biochemically and phenotypically dissected into two subcomplexes: a trimer of Vps35, Vps29, and Vps26 (referred to as CSC) that mediates cargo selection, and a dimer of Vps5 with Vps17 (referred to as SNX) that promotes tubule or vesicle formation (Seaman et al., 1998; Seaman, 2004, 2012). In addition to the recycling function of endosomal cargo, Retromer interacts with yeast PROPPIN Atg18 to form the CROP complex, which mediates vacuole membrane fission (Courtellemont et al., 2022). Since Any1 is an endosomal protein and is also present on the Golgi, we wondered whether Any1 is a Retromer cargo that recycles from the EE to the Golgi. We therefore analyzed the localization of Any1-mNeon in wild-type, Retromer mutants (vps35Δ, vps5Δ), and a CROP mutant (atg18Δ). In vps35Δ or vps5Δ cells, Any1 was mostly missorted in the vacuole lumen, whereas the atg18Δ mutant had no effect on Any1 localization (Fig. 1 F). These data suggest that Any1 cycles between the Golgi and endosome with the help of the retromer complex. To determine whether Any1 dynamically localizes to

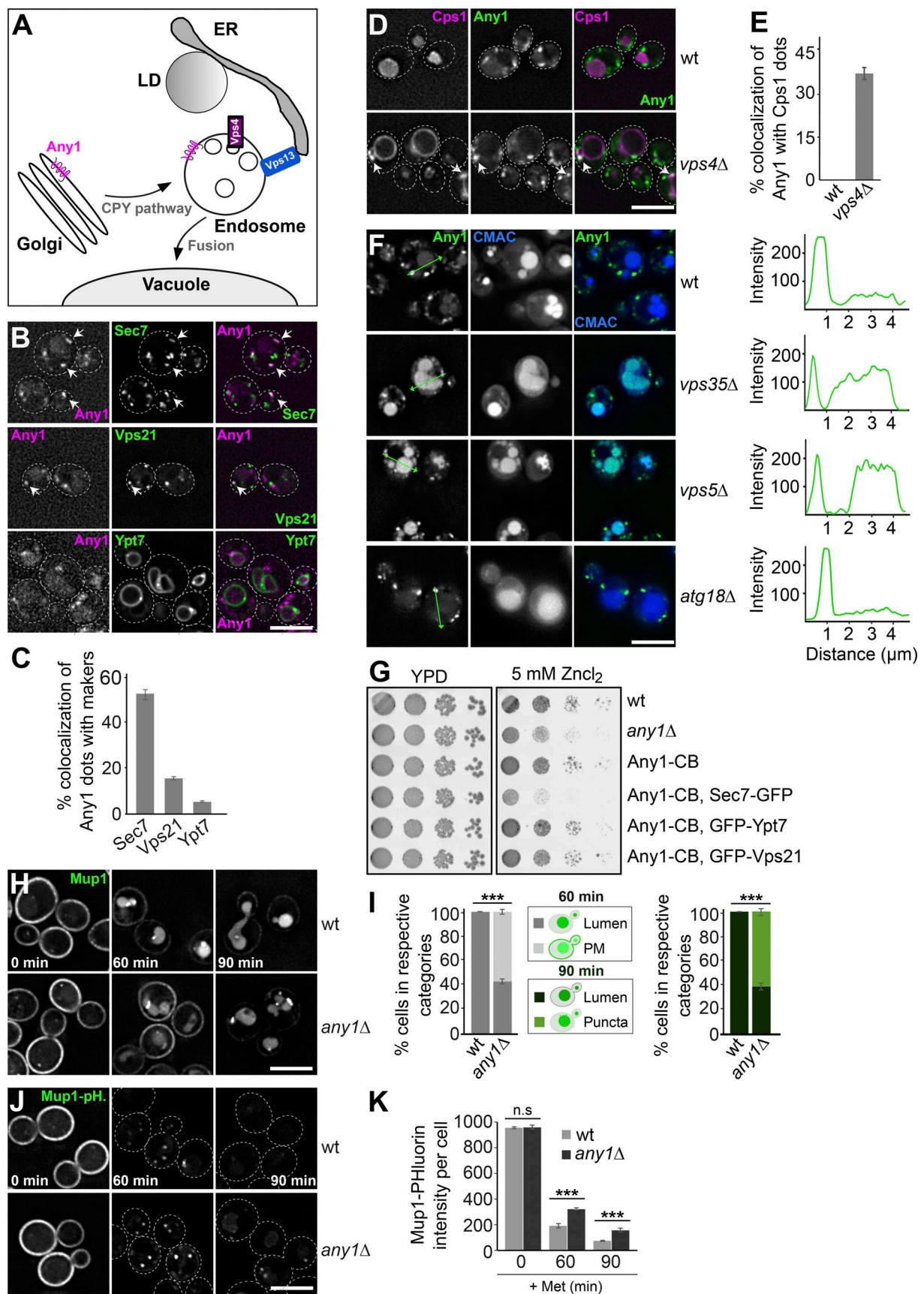

Figure 1. **Any1 cycles between the Golgi and early endosomes. (A)** Working model of Any1 trafficking. For details see text. **(B)** Localization of Any1 relative to Sec7, Vps21, and Ypt7. Cells expressing mKate-tagged Any1 and GFP-tagged Sec7, Vps21, or Ypt7 were grown in a synthetic medium and analyzed by

fluorescence microscopy, and individual slices are shown. Scale bar, 5 µm. **(C)** The percentage of Any1 puncta colocalizing with Sec7, Vps21, or Ypt7 was determined. Any1 dots ($n \geq 600$), Sec7 dots ($n \geq 900$), Vps21 dots ($n \geq 600$), and Ypt7 dots ($n \geq 200$) were quantified using a custom graphical user interface based on ImageJ built-in routines (Arlt et al., 2015; Gao et al., 2018). Single cells were segmented from bright-field images, and protein dot signals were identified on sum projections of deconvolved image stacks. Colocalization was assessed by counting particles with >50% overlapping area, with manual corrections to reduce background artifacts. Error bars represent standard deviation (SD). Bar graphs represent averages from three independent experiments. **(D)** Localization of Any1 relative to Cps1 in wild-type and *vps4Δ* cells. Cells expressing mNeon-tagged Any1 and mCherry-tagged Cps1 were grown in a synthetic medium and analyzed by fluorescence microscopy, and individual slices are shown. Scale bar, 5 µm. **(E)** Percentage of Any1 puncta colocalizing with Cps1 dots. Any1 dots ($n \geq 500$) and Cps1 dots ($n \geq 80$) were quantified by Image J as described in Fig. 1 C. Error bars represent standard deviation (SD). Bar graphs represent averages from three independent experiments. **(F)** Localization of Any1 in wild-type, *vps35Δ*, *vps5Δ*, and *atg18Δ* cells. Cells expressing mNeon-tagged Any1 were grown in a synthetic medium and vacuoles were stained with CMAC. Cells were analyzed by fluorescence microscopy and individual slices are shown. Scale bar, 5 µm. The line profiles on the right show the fluorescence of Any1-mNeon signal along the green lines indicated in the images. **(G)** Growth assay on ZnCl$_2$-containing plates. The indicated cells were grown in synthetic medium, spotted onto plates containing YPD with or without 5 mM Zn$^{2+}$, and grown at 30°C for 2–3 days. **(H)** The sorting of Mup1-GFP in wild-type and *any1Δ* cells. Cells expressing Mup1-GFP were grown in the absence of methionine (0 min) in a minimal medium to an OD$_{600}$ of 0.8. Then methionine was added, and cells were analyzed by fluorescence microscopy after 60 or 90 min. The individual slices are shown. Scale bar, 5 µm. **(I)** Quantification of Mup1-GFP sorting from H. Cells ($n \geq 150$) were quantified by Image J. Bar graphs represent averages from three independent experiments. Error bars represent standard deviation (SD). ***, $P < 0.001$ (two-tailed Student's $t$ test). **(J)** Sorting of Mup1-pHluorin in wild-type and *any1Δ* cells. Cells were analyzed under the same growth conditions and procedures as described in Fig. 1 H. **(K)** Quantification of Mup1-pHluorin intensity per cell. Cells ($n \geq 150$) were quantified by ImageJ. The analysis involved adjusting the threshold, creating a mask, and measuring the intensity within selected areas. Bar graphs represent averages from three independent experiments. Error bars represent standard deviation (SD). ns, > 0.05; ***, $P < 0.001$ (two-tailed Student's $t$ test).

---

both Golgi and endosomes to control its activity, we tagged the Golgi marker protein Sec7, early endosome marker protein Vps21, or the LE marker protein Ypt7 with GFP in strains expressing endogenously tagged Any1 carrying a nanobody against GFP (chromobody, CB). We used this approach successfully to confine proteins at specific subcellular locations (Malia et al., 2018; Füllbrunn et al., 2024). However, Any1 is a transmembrane protein, whereas other protein markers are membrane-associated proteins. This raises the possibility that Any1 in turn recruits marker proteins to its specific location. However, cells expressing Any1-CB did not alter the localization of GFP-tagged Sec7, Vps21, or Ypt7 relative to the vacuole, nor did it affect the number of Vps21 or Ypt7 puncta (Fig. S1, A and B), but only reduced the number of Sec7 puncta (Fig. S1 C). This observation suggests that Any1 is largely recruited by the GFP-tagged marker proteins to specific subcellular locations. We then analyzed Any1-CB cells and cells lacking Any1 in the growth assay. As previously observed (Fig. S1 A), the *any1Δ* mutant exhibited a significant growth defect, which was even more pronounced in the cells expressing Any1-CB with Sec7-GFP. Interestingly, cells expressing Any1-CB with GFP-Vps21 or GFP-Ypt7 had no growth defect (Fig. 1 G), indicating that Any1 may function at endosomes. To further analyze the role of Any1 at endosomes, we followed the transport of the PM-localized methionine permease Mup1 (Menant et al., 2006). In the absence of methionine, Mup1 localizes to the PM. Once methionine is added, Mup1 is endocytosed and sorted into ILVs at the endosome by ESCRTs. To evaluate Mup1 sorting, we expressed a GFP-fused Mup1 in wild-type and *any1Δ* cells, where Mup1-GFP localized to the PM in the absence of methionine. Upon addition of methionine for 90 min, Mup1-GFP was completely sorted into the vacuole lumen in wild-type cells (Fig. 1 H). In contrast, Mup1-GFP sorting to the vacuole lumen was strongly reduced in *any1Δ* cells and some protein accumulated in punctate structures next to the vacuoles, indicating that Any1 function supports Mup1 sorting (Fig. 1, H and I). To strengthen this observation, we used a pH-sensitive GFP variant, pHluorin, fused to Mup1 (Miesenböck et al., 1998). When Mup1–pHluorin is internalized into ILVs and delivered to

the vacuole, the acidic environment quenches the fluorescence, enabling us to assess the efficiency of cargo sorting at the endosome and its delivery to the vacuole. By 90 min of methionine, the pHluorin fluorescence was largely quenched in wild-type cells, confirming the efficient sorting of Mup1–pHluorin into ILVs and delivery to the vacuole (Fig. 1, J and K). In contrast, the pHluorin signal in *any1Δ* cells remained visible at the endosome even after 90 min of methionine addition (Fig. 1, J and K), indicating that Any1 is required for efficient endolysosomal transport toward the vacuole.

## Any1 is a lipid scramblase

Any1 has been suggested to function antagonistically to phospholipid flippases and may thus act as a scramblase on the endosomes or the Golgi (Yamamoto et al., 2017; van Leeuwen et al., 2016). To test this hypothesis, we first carried out structural predictions with AlphaFold2 (AF2) since Any1 structure has not been experimentally determined (Fig. 2 A). The predicted model for the monomer of Any1 consists of a luminal N-terminal helix (H1), followed by three transmembrane (TM) helices (T1, T2, and T3) and a cytosolic domain that includes three helices (H2, H3, and H4). After this cytosolic domain, four TM helices are found (T4, T5, T6, and T7), with T5 containing a PQ motif (Pro270 and Gln271). The model ends with one last helix (H5) at the C-terminus. A similar topology (N-terminal at the lumen, seven TM helices, and C-terminal at the cytosol) has been described for other PQ-loop proteins such as Cystinosin and Sweet transport in humans (Guo et al., 2022) and plants (Löbel et al., 2022; Han et al., 2017; Tao et al., 2015). Interestingly, according to AF2, Any1 and its homolog human protein, PQLC1 (SLC66A2), share a very similar fold for their TM structures (Fig. 2 B), with a root-mean-square-deviation (RMSD) between their C-alpha of only 1.107 Å (for a total of 120 residues). This structural similarity suggests that SLC66A2 may have conserved function to yeast Any1.

Next, to directly test the scramblase activity of Any1, we first purified Any1 from yeast cells and analyzed it by size exclusion chromatography (SEC). Surprisingly, the size of Any1 after SEC

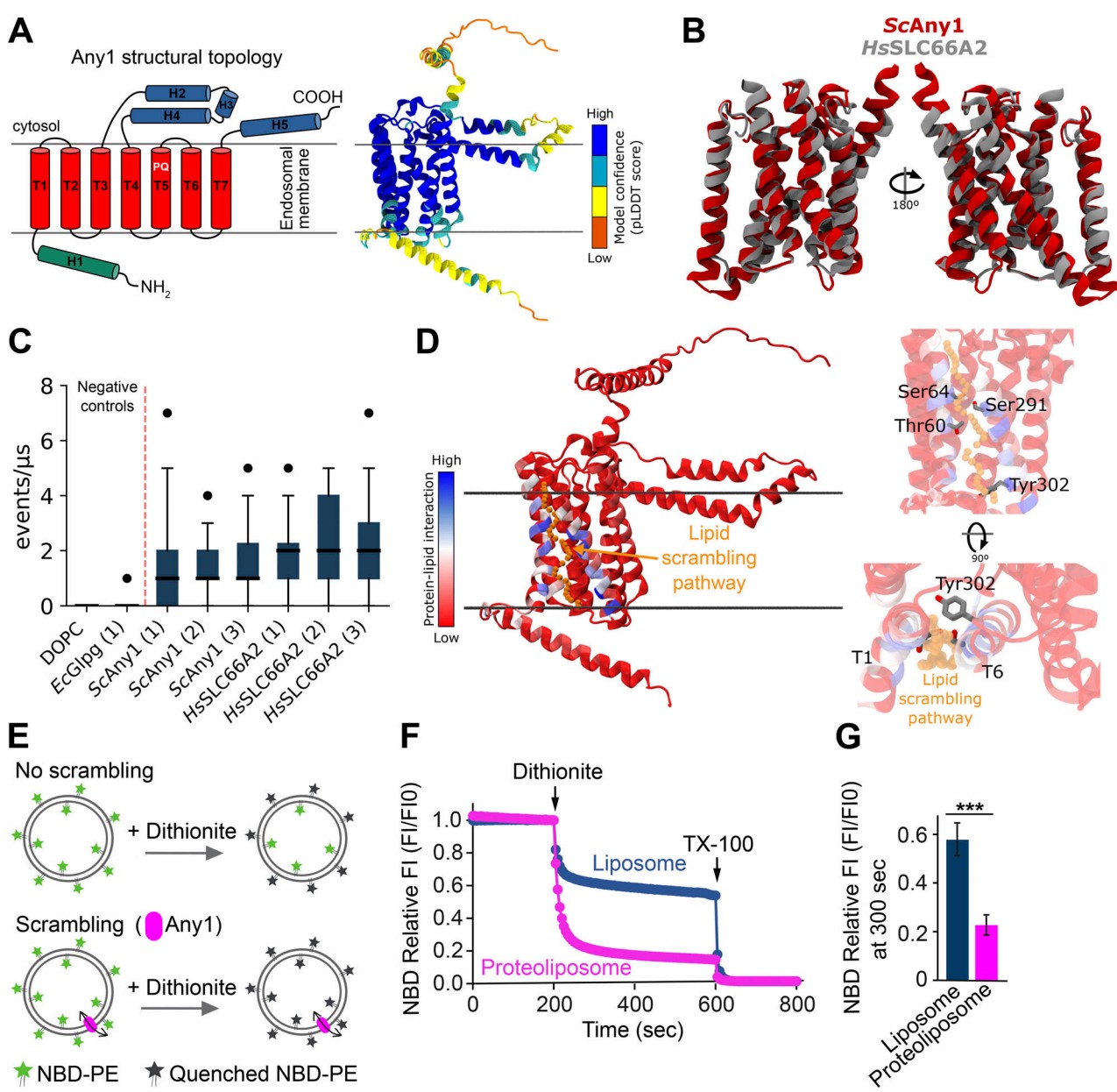

Figure 2. **Any1 has phospholipid scramblase activity in silico and in vitro. (A)**. Left: Secondary structure topology of the yeast Any1 (*Sc*Any1) protein. The luminal, transmembrane, and cytosolic helices are colored in green, red, and blue, respectively; The PQ-loop motif is highlighted. Right: Predicted structural model by AlphaFold2 for the ScAny1 protein colored by its confidence prediction. **(B)** Structural alignment of the transmembrane segments of *Sc*Any1 (red) and *Hs*SLC66A2 (gray) proteins. **(C)** In silico lipid scrambling assay from CG-MD simulations for different oligomerizations (number in parentheses) of both *Sc*Any1 and *Hs*SLC66A2. The values for the negative controls (DOPC pure membrane and EcGlpG) are taken from Li et al. (2024a). The blue boxes represent the interquartile range while the solid black line inside each box represents the median of the data. The minimum and maximum data are shown with whiskers and outliers are shown as black dots. Each data point represents the total sum of events that occurred per microsecond as mentioned in Materials and methods. Each boxplot contains 36 data points corresponding to 18 µs production runs over two replicas after omitting the first 2 µs for equilibration from each trajectory. **(D)** Main amino acids involved in the in silico lipid scrambling activity of *Sc*Any1. Blue indicates a high contribution to lipid scrambling, red indicates no contribution to lipid scrambling. Right, top, and side views of the main polar residues involved in the lipid scrambling. **(E–G)** Fluorescence-based lipid scramblase assay of Any1. **(E)** Schematic representation of the scramblase assay. **(F)** NBD relative fluorescence during the scrambling assay (details of the assay are described in Materials and methods). **(G)** Quantification of fluorescent signal from F. Bar graphs represent averages from three independent experiments. Error bars represent SD of three independent experiments. ***, P < 0.001 (two-tailed Student's *t* test).

was around 158 kDa, while Any1 has a predicted molecular weight of 46.8 kDa (Fig. S2, A and B), suggesting that Any1 might exhibit different oligomeric states, potentially forming a homotrimer or dimer. Using a well-established in silico scrambling assay based on unbiased coarse-grain (CG) molecular dynamics

(MD) simulations (Li et al., 2024a), we examined the scramblase activity of multiple oligomeric assemblies (monomer, dimer, trimer) of Any1 and SLC66A2, as predicted by AF2 (Fig. S2 C). Our in silico assay shows that both Any1 and SLC66A2 can scramble lipids regardless of their oligomerization state (Fig. 2 C).

The in silico lipid scrambling activity obtained by Any1 and SLC66A2 was comparable with other scramblases previously studied through similar approaches and supported with in vitro assays (Bushell et al., 2019; Jahn et al., 2023; Li et al., 2024a). Analysis of the simulations suggests that the main lipid-scrambling pathway for Any1 is located at the interface between TM helices T1 and T6 (Fig. 2 D). In this region, four polar amino acids (T60, S64, S291, and Y302) form a polar region at the protein–membrane interface and are highly involved in lipid scrambling. Additionally, since our SEC results suggest Any1 might preferentially form a homotrimer, further analysis of the homotrimer Any1 simulations shows that the Any1 trimer has the ability to induce extensive membrane curvature in its proximity (Fig. S2 D).

To investigate whether Any1 is able to transport lipids from the inner to the outer leaflet of liposomes in vitro, we used a previously established in vitro scramblase assay (Chumpen Ramirez et al., 2023) (Fig. 2 E). In this assay, the fluorescent lipid NBD-PE is reconstituted into liposomes lacking or containing purified candidate proteins. After reconstitution of proteoliposomes, NBD-PE is present in both the inner and outer leaflets. Proteins with scramblase activity will continuously transfer lipids between both leaflets. By treatment of a membrane-impermeable reducing agent dithionite, the fluorescence of NBD-PE within the outer liposome leaflet is reduced and irreversibly quenched, thus resulting in an ∼50% loss of fluorescence in the absence of a scramblase. In contrast, the addition of Triton X-100 allows dithionite to access and quench the fluorescence of NBD-PE within both leaflets. Likewise, the presence of a scramblase will lead to the quenching of NBD-PE in both leaflets, as lipids continuously equilibrate across the bilayer. When we recorded the fluorescence of our liposomes, the protein-free liposomes showed indeed a reduction of nearly 50% after adding dithionite, as expected (Fig. 2 F). In contrast, liposomes containing Any1 showed a further reduction in the fluorescence of the NBD, indicating that the NBD-PE was scrambled between both leaflets (Fig. 2, F and G). To determine that the increased reduction of the NBD by dithionite is due to the enzymatic activity of Any1 and not increased liposome permeability, we performed a fluorescence protection assay. For this, we analyzed the accessibility of the dithionite to soluble 2-NBD-glucose (2-NBDG), a fluorescent analog of glucose, which was trapped inside liposomes. We measured the fluorescence of 2-NBDG over time and observed that most of the signal remained unchanged after the addition of dithionite. We observed a rapid and weak reduction of fluorescence upon the addition of dithionite, which was due to quenching of some of the 2-NBDG remaining in the liposome solution after preparation. Importantly, upon the addition of Triton X-100, the fluorescence of the captured 2-NBDG was quenched to background levels, suggesting that the dithionite could not enter the interior of the liposomes during the measurement unless the liposomes were lysed (Fig. S2 E). Taken together, our data reveal that Any1 is a phospholipid scramblase.

To investigate whether the predicted polar region at the protein–membrane interface is required for lipid scrambling, we mutated T60, S64, and Y302 in Any1 to leucine and assessed its

scramblase activity in silico. Interestingly, Any1$^{T60L}$, Any1$^{S64L}$, and the double mutant Any1$^{T60L,S64L}$ significantly lost scramblase activity in silico (Fig. 3 A), whereas Any1$^{Y302L}$ did not. These findings suggest that T60 and S64 are likely critical for Any1's scramblase function. To further explore the functional importance of these two residues for Any1 function, we analyzed the localization of the Any1$^{T60L,S64L}$ mutant protein relative to Sec7-marked trans-Golgi network, Vps21-labeled EE, and Ypt7-marked LE (as in Fig. 1 B). The Any1$^{T60L,S64L}$ mutant protein localized normally to the Golgi, but was more abundant on EE compared with wild-type cells (Fig. 3, B, C, and G). Strikingly, these mutations in Any1 caused an accumulation of Vps21-positive EE, reduced the number of Ypt7-positive LE, and slightly disrupted the Sec7-marked trans-Golgi network (Fig. 3, B–F). Taken together, these findings imply that T60 and S64 are critical for Any1 function.

## LDs are in close proximity to endosomes

Our data suggest that Any1 is a lipid scramblase that localizes to EE, where it is required for endosomal function. We wondered if Any1 functions like Atg9 on autophagosomes, which cooperates with the lipid transfer protein Atg2 in phagophore expansion (Ghanbarpour et al., 2021; Chumpen Ramirez et al., 2023; van Vliet et al., 2022; Matoba et al., 2020). To determine a possible function of Any1 and to identify a possible partner lipid transfer protein at endosomal MCS, we first analyzed such MCSs by transmission electron microscopy (TEM) in yeast cells. To achieve the best preservation of membrane compartments and avoid possible artifacts due to chemical fixation, we used the high-pressure freezing (HPF) method. Here, samples are vitrified under very high pressure in liquid nitrogen, followed by freeze substitution (FS). In wild-type cells, we observed the well-maintained ultrastructure of both the normal MVBs and class E compartments consisting of a curved stack of cisternal membranes in vps4Δ cells (Fig. S3, A–C). This showed that our HPF and FS protocol maintained the ultrastructure of organelles in yeast cells.

At MCSs, membranes of two different organelles come into close apposition (10–40 nm). We therefore analyzed MVBs as organelles with a clear morphology and identified them predominantly adjacent to the vacuoles (Fig. 4, A and B). In line with previous studies, MVBs also established MCSs with the ER to facilitate MVB biogenesis (Suzuki et al., 2024) (Fig. 4 B; and Fig. S3, E and F) and were found close to mitochondria as described before (Todkar et al., 2019) (Fig. 4 B and Fig. S3 G). However, MVBs barely formed MCSs with LDs (Fig. 4 B and Fig. S3 D).

We further analyzed MCSs between the intermediate endosome structures and other organelles in vps4 temperature-sensitive (ts) mutant by TEM. In yeast, prolonged inactivation of this mutant at a restrictive temperature (37°C) results in the formation of class E compartments (Babst et al., 1998; Adell et al., 2017). Therefore, we analyzed MCSs of the intermediate endosomal structures with other organelles at different time points at the restrictive temperature. Endosomes in vps4ts cells appeared unperturbed at permissive temperature (23°C) and comparable with MVBs in wild-type cells (Fig. 4 C). However, the number of ILVs decreased dramatically within

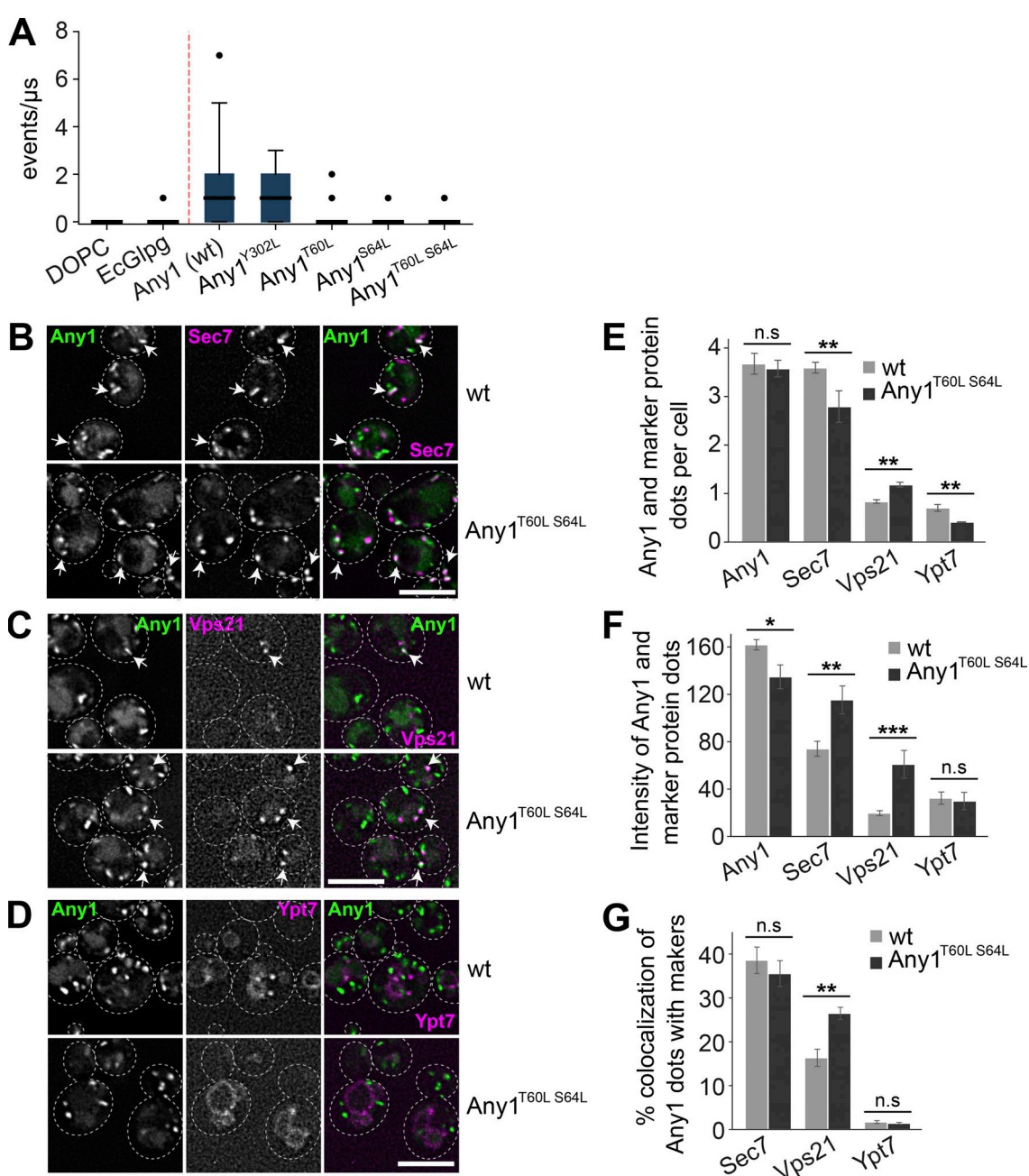

Figure 3. **The predicted scrambling pathway is critical for Any function. (A)** In silico lipid scrambling assay of Any1 mutants. The values for the negative controls (DOPC pure membrane and EcGlpG) are taken from Li et al. (2024a). The blue boxes represent the interquartile range while the solid black line inside each box represents the median of the data. The minimum and maximum data are shown with whiskers and outliers are shown as black dots. Each data point represents the total sum of events that occurred per microsecond as mentioned in Materials and methods. Each boxplot contains 36 data points corresponding to 18-μs production runs over two replicas, after omitting the first 2-μs for equilibration from each trajectory. **(B–D)** Localization of Any1 wild-type and mutant relative to Sec7, Vps21, and Ypt7. Plasmids encoding for Any1-mNeon wild-type and mutant proteins under the control of its endogenous promoter were expressed in any1Δ cells. These cells also expressed mScarlet-tagged Sec7, mCherry-tagged Vps21, or Ypt7 and were grown in a synthetic medium. They were analyzed by fluorescence microscopy. Images show individual slices. Scale bar, 5 μm. **(E and F)** Quantification of the number and intensity of Sec7, Vp21 and Ypt7 dots. The number of dots was counted manually from a sum projection. The intensity analysis of fluorescent dots involved adjusting the threshold, creating a mask, and measuring the intensity within selected areas. Bar graphs represent averages from three independent experiments. Error bars represent standard deviation (SD). ns, > 0.05; **, P < 0.01, ***, P < 0.001 (two-tailed Student's t test). **(G)** Percentage of Any1 puncta colocalizing with Sec7, Vps21, or Ypt7 was determined. Any1 dots (n ≥ 600), Sec7 dots (n ≥ 600), Vps21 dots (n ≥ 200), and Ypt7 dots (n ≥ 100) were quantified using a custom graphical user interface based on ImageJ built-in routines as described in Fig. 1 C. Bar graphs represent averages from three independent experiments. ns, > 0.05; **, P < 0.01 (two-tailed Student's t test).

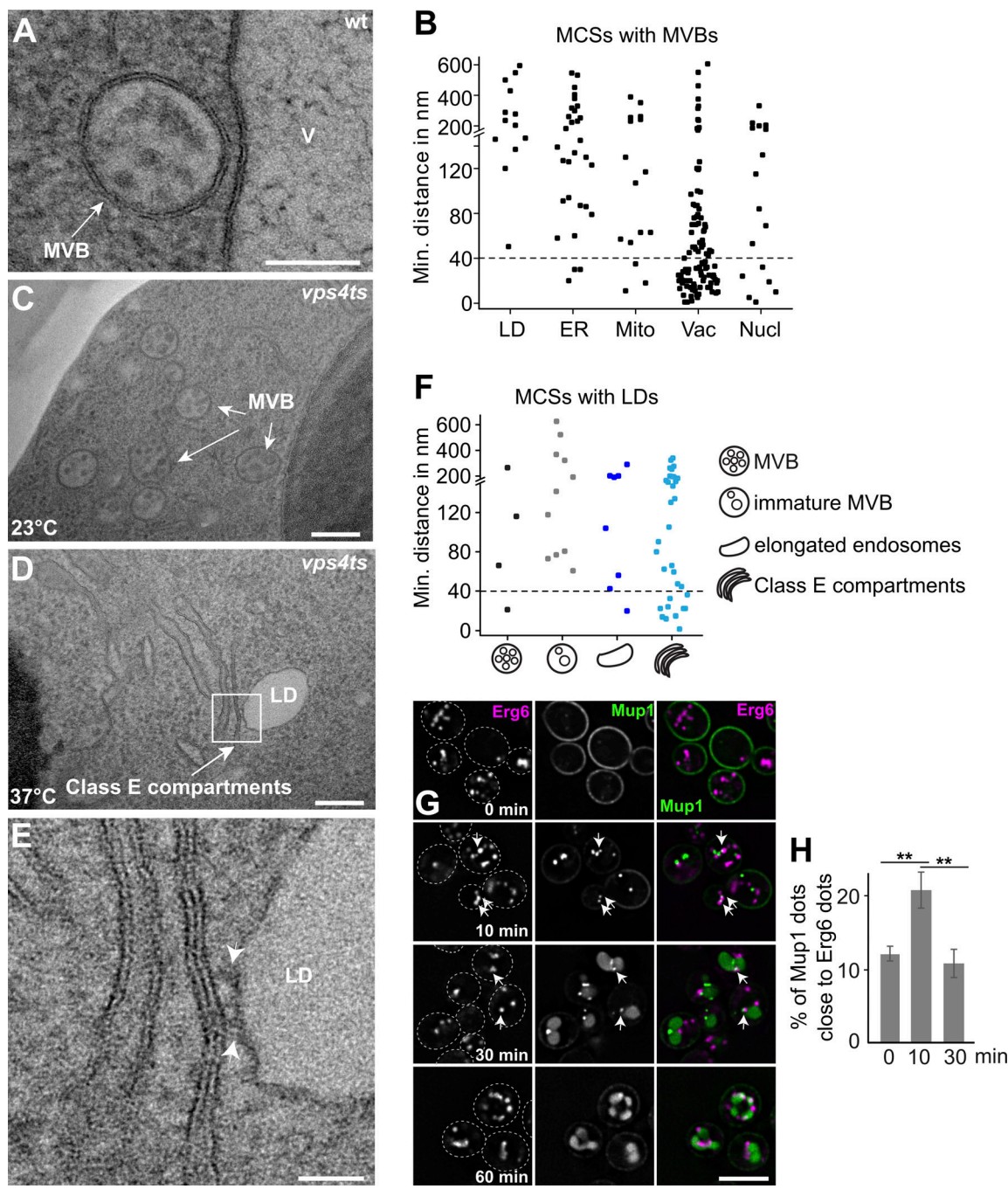

Figure 4. **Lipid droplets are in close proximity to endosomes. (A)** Analysis of MCSs between the endosomes and LD or ER or mitochondria or vacuoles or nucleus by transmission electron microscopy in wild-type cells. Scale bar, 100 nm. **(B)** Quantification of the distance between LDs or ER or mitochondria (Mito) or vacuoles (Vac) or nuclei (Nucl) from A ($n$ = 118 cells). **(C–E)** Analysis of MCSs between LDs and endosomal structures in *vps4 ts* mutant cells. *vps4 ts* cells were grown at 23°C in a synthetic medium and then shifted or not to 37°C for 2 h. Scale bars, 200 nm (C and D), 50 nm (E). **(F)** Quantification of the distance between LDs and the intermediate endosomal structures in *vps4 ts* cells at 37°C for 2 h ($n$ = 87). **(G)** Localization of Mup1 relative to Erg6. Cells expressing GFP-tagged Mup1 and mKate-tagged Erg6 were grown in the absence of methionine (0 min) in a minimal medium to an $OD_{600}$ of 0.8. Then methionine was added and cells were analyzed by fluorescence microscopy after 10, 30, and 60 min. Scale bar, 5 μm. **(H)** A fraction of Mup1 puncta partially overlaps with Erg6 dots. Mup1 dots ($n$ ≥ 200) were quantified by Image J. Quantification was conducted as described in Fig. 1 C. The proximity between Erg6 and Mup1 dots was evaluated by counting particles with >10% overlapping area. Bar graphs represent averages from three independent experiments. Error bars represent standard deviation (SD). **, P < 0.01 (two-tailed Student's $t$ test).

30 min at 37°C, coinciding with a significant increase of several nonspherical and flattened endosomes (Fig. S4, A and B). Over time, endosomes accumulated and appeared more extensively flattened, resulting eventually in a typical class E compartment morphology after 60 min of Vps4 inactivation (Fig. 4 D; and Fig. S4, A and B).

Interestingly, we observed an increasing number of LDs proximal to Class E compartments, whereas LDs were barely found next to intermediate endosomal structures (Fig. 4, D–F and Fig. S4, C–F) or MVBs (Fig. 4 B). Importantly, the number of LDs did not change during inactivation of Vps4 (Fig. S4 G). In contrast, the vacuole remains in contact with the Class E compartment as expected, whereas the ER and mitochondria form only a few contact sites with Class E compartments. These data suggest that LDs may establish MCSs with endosomal structures to promote their biogenesis.

To determine if LD–endosome contact sites form during endosome biogenesis, we induced endocytosis of PM-localized Mup1-GFP (Fig. 1 H) in cells co-expressing Erg6-mKate as an LD marker. We then determined the localization of Erg6-mKate marked LDs relative to Mup1-GFP positive endosomes over the time course of endocytosis. Notably, 10 min after the addition of methionine, Erg6-mKate accumulated next to Mup1-GFP, which decreased after 30 min (Fig. 4, G and H). This suggests that LDs are in close proximity to endosomes and might form transient MCSs with endosomes. In agreement, we noticed that only small puncta of Mup1-labeled endosomes contacted LDs (Fig. 4 G), suggesting that only EE are proximal to LDs.

### LDs and Any1 are required for endosome biogenesis

LDs are ubiquitous organelles that originate from the ER. They consist of a neutral lipid core predominantly formed by triacylglycerols (TAGs) and sterol esters (SEs), surrounded by a phospholipid monolayer (Walther et al., 2017). To further examine if the potential LD-endosome contact sites are involved in MVB biogenesis, we first asked whether LDs play a crucial role in the endocytic pathway. To test this, we analyzed the sorting of mCherry-Cps1 in a yeast strain lacking the four proteins required for TAG and SE synthesis: Dga1, Lro1, Are1, and Are2 (are1Δare2Δdga1Δlro1Δ; LDΔ), which results in a loss of LDs and compared those with wild-type and vps4Δ mutant cells after Bodipy staining to mark LDs. Whereas mCherry-Cps1 was found exclusively in the vacuolar lumen in wild-type cells, mCherry-Cps1 accumulated in the vacuolar membrane and less so inside the vacuole in LDΔ cells (Fig. 5, A and C). Upon deletion of VPS4, Cps1 accumulated in the vacuolar membrane and Class E compartments (Fig. 1 D and Fig. 5 B). Intriguingly, in LDΔvps4Δ cells, mCherry-Cps1 did not cluster at Class E compartments but was found only at the vacuolar membrane (Fig. 5, D and J). Taken together, these data suggest that the endocytic pathway requires LDs by potentially promoting the formation of intermediate endosomal structures. Surprisingly, Cps1 sorting to the vacuole lumen was not disrupted in any1Δ cells possibly because we monitored the entire pool of Cps1 and thus predominantly the vacuole rather than its sorting from the Golgi to the vacuole. Additionally, Cps1 is delivered to maturing endosomes, whereas Any1 function may be required at an earlier stage in endosome biogenesis (Fig. 5, E and J). However, when cells lacked both Any1 and Vps4, mCherry-Cps1 also lost its localization to the Class E compartment (Fig. 5, F and J), a defect similarly observed in LDΔvps4Δ cells. This indicates that Any1 and LDs may function in the same pathway involved in endosomal maturation. We next asked which lipid transfer protein is present at these

potential LD–endosome contact sites. Given that lipid transfer protein Vps13 has been suggested to localize to ER–endosome contact sites and is required for endosome biogenesis (Suzuki et al., 2024), we wondered whether Vps13 also plays an essential role at these potential LD–endosome contact sites. To test this, we examined if Vps13 is required for the endosomal function by analyzing the sorting of mCherry-Cps1 (Fig. 5, A, B, and G) or Mup1-GFP (Fig. 5 L) in wild-type, vps4Δ, and vps13Δ cells. As in LDΔ cells, mCherry-Cps1 accumulated in the vacuolar membrane in vps13Δ cells (Fig. 5 G), which was similar to the transport of Mup1-GFP after 90 min of methionine addition (Fig. 5, L and M), in agreement with recent findings (Suzuki et al., 2024). Further, Ypt35 recruits Vps13 to endosomal and vacuolar membranes (Bean et al., 2018). In vps13Δvps4Δ and ypt35Δvps4Δ cells, we surprisingly noticed the massive accumulation of mCherry-Cps1 in special membrane clusters, which were distinct from Class E compartments (Fig. 5, H–J). This indicates that Class E compartments may not form in these mutants. Taken together, our data confirmed that Vps13 is indeed required for the endocytic pathway, likely to provide lipids at MCSs to facilitate endosome formation. Additionally, our analysis of various mutants revealed no significant differences in the overall number of LDs (Fig. 5, A, B, E–I, and K), indicating that Any1 and Vps13 are not essential for LD function.

### Vps13 supports crosstalk between LDs and endosomes

Since Vps13 is critical for endosomal function and is required for endosome biogenesis (Suzuki et al., 2024) (Fig. 5, A, B, and G–J), we wondered whether Vps13 cooperates with Any1 at the potential LD–endosome MCSs, similar to the cooperation of Atg2 and Atg9 in phagophore expansion (Chumpen Ramirez et al., 2023). To test this hypothesis, we first analyzed the colocalization of Vps13-mNeon with Any1-mKate in wild-type and vps4Δ cells. As Vps13 functions at various MCSs (Dziurdzik and Conibear, 2021; Ugur et al., 2020; Melia and Reinisch, 2022), it only partially colocalized with Any1 in wild-type cells. However, in vps4Δ cells, colocalization of Vps13 with Any1 was clearly increased, likely due to the fact that both are endosomal proteins (Fig. 6, A and B).

To further substantiate that Vps13 localizes to MCSs between LDs and endosomes, we monitored the localization of Vps13-mNeon relative to Faa4-marked LDs in cells that were incubated with Cy5-labeled α-factor (Fig. 6 C). At the PM, α-factor binds to the pheromone receptor Ste2 and is then endocytosed and transported to the vacuole lumen for degradation, which can be monitored over time (Gao et al., 2022). We observed that Vps13 colocalized with α-factor over time. Some of these colocalizing dots were proximal to Faa4, starting at 5 min, again suggesting that Vps13 is present at endosomes and in close proximity to LDs (Fig. 6, C and D).

Since Vps13 may mediate lipid transfer at the ER–endosome contact sites (Suzuki et al., 2024), we asked whether Vps13 is critical for facilitating both ER–endosome contact sites and potential LD–endosome contact sites. To investigate this, we traced the colocalization of Erg6-mKate marked LDs or Sec63-mCherry marked ER with Mup1-GFP labeled endosome over the time course in both wild-type and vps13Δ cells. In wild-type cells,

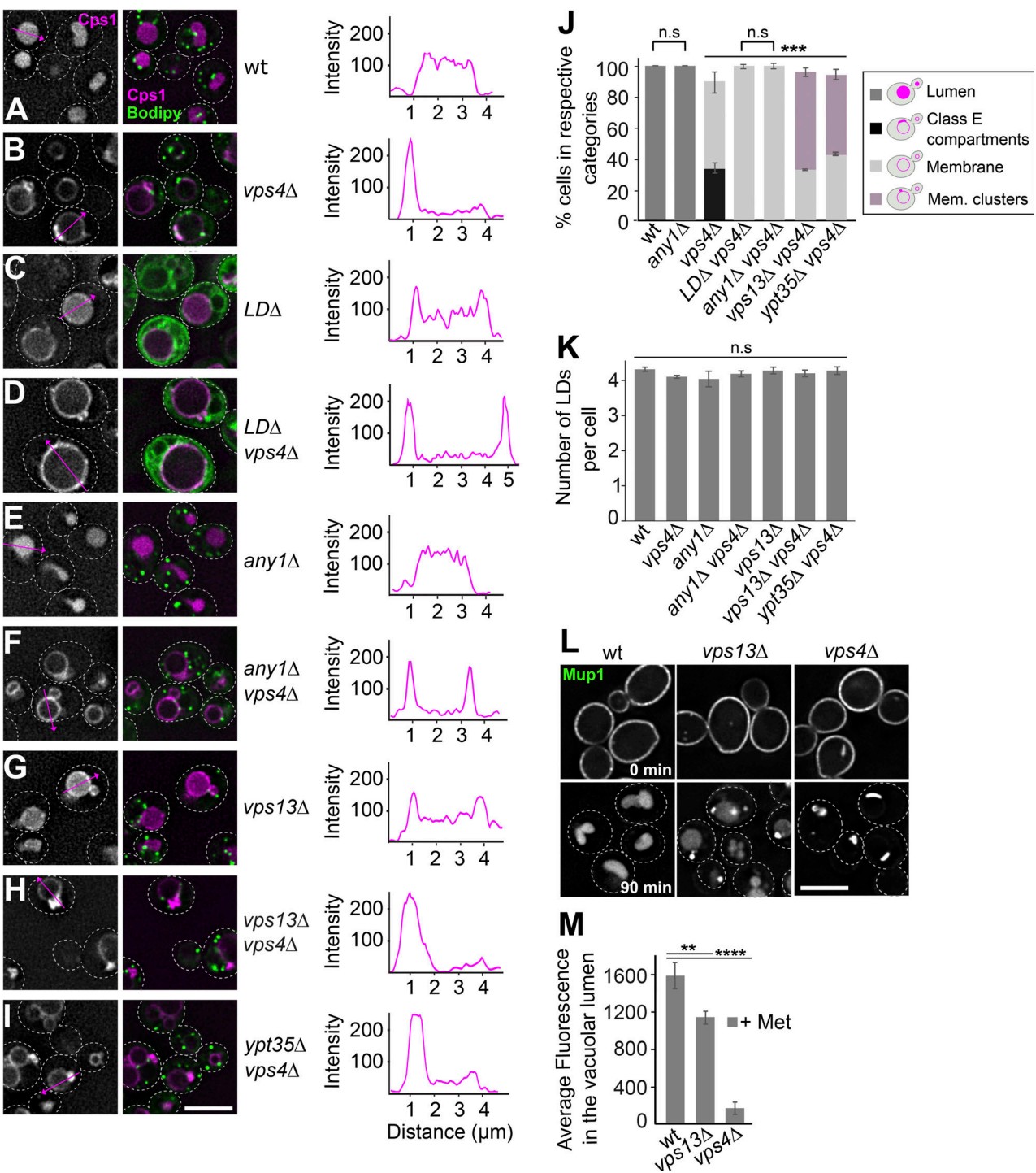

Figure 5. **LDs and Vps13 are important for efficient endolysosomal protein trafficking. (A–I)** Sorting of Cps1 in wild-type and mutant cells. Cells expressing mCherry-tagged Cps1 were grown in a synthetic medium with LDs stained with Bodipy 493/503 (green) and analyzed by fluorescence microscopy. Representative slides are shown. Scale bar, 5 µm. The line profiles on the right show the fluorescence of mCherry-Cps1 signal along the magenta lines indicated in the images. **(J)** Quantification of Cps1 sorting from A–I. Cells ($n \geq 200$) were quantified by Image J. Bar graphs represent averages from three independent experiments. Error bars represent standard deviation (SD). ns, > 0.05; ***, P < 0.001 (two-tailed Student's $t$ test). **(K)** Quantification of the number of LDs from A, B, and E–I. Cells ($n \geq 200$) were quantified by ImageJ. The number of LDs was counted manually from a sum projection. Bar graphs represent averages from three independent experiments. Error bars represent standard deviation (SD). ns, > 0.05 (two-tailed Student's $t$ test). **(L)** Sorting of Mup1 in wild-type, $vps13\Delta$, and $vps4\Delta$ cells. Cells expressing GFP-tagged Mup1 in the absence of methionine (0 min) in minimal medium to an $OD_{600}$ of 0.8. Methionine was then added and cells were analyzed by fluorescence microscopy after 90 min. Scale bar, 5 µm. **(M)** Quantification of Mup1 sorting from J. Cells ($n \geq 100$) were quantified by Image J using the region of interest manager tool. Bar graphs represent averages from three independent experiments. Error bars represent standard deviation (SD). **, P < 0.01; ***, P < 0.001 (two-tailed Student's $t$ test).

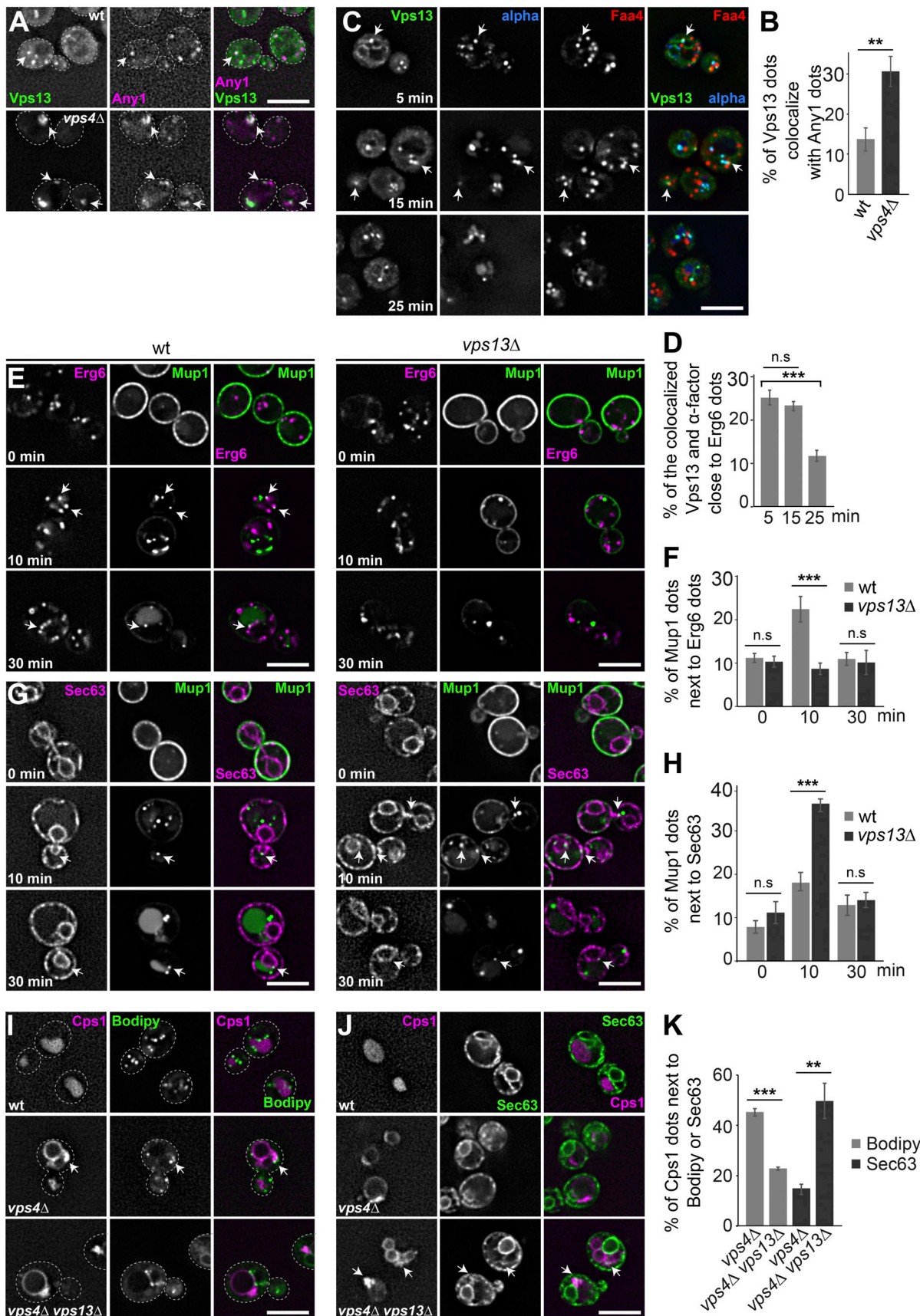

Figure 6. **Vps13 supports crosstalk between the endosomes and LDs. (A)** Localization of Vps13 relative to Any1 in wild-type and *vps4Δ* cells. Cells expressing mNeon-tagged Vps13 and mKate-tagged Any1 were grown in a synthetic medium and analyzed by fluorescence microscopy, and individual slides are

shown. Scale bar, 5 µm. **(B)** Percentage of Vps13 puncta colocalizing with Any1 dots. Vps13 dots ($n \geq 150$) and Any1 dots ($n \geq 300$) were quantified by Image J as described in Fig. 1 C. Bar graphs represent averages from three independent experiments. Error bars represent standard deviation (SD). **, P < 0.01 (two-tailed Student's $t$ test). **(C)** Localization of Vps13 relative to α-factor and Faa4. Cells expressing mNeon-tagged Vps13 and mScarlet-tagged Faa4 were grown in a synthetic medium, cooled to 4°C to block endocytosis, and treated with fluorescent α-factor for 15 min at 4°C. Cells were analyzed by fluorescence microscopy after being shifted to 23°C for the indicated time points. Individual slides are shown. Scale bar, 5 µm. **(D)** Percentage of the colocalized Vps13 and α-factor dots partially overlapped with Erg6 dots. Vps13 dots ($n \geq 400$) and α-factor dots ($n \geq 200$) were quantified by Image J as described in Fig. 1 C and Fig. 3 H. Bar graphs represent averages from three independent experiments. Error bars represent standard deviation (SD). ns, > 0.05; ***, P < 0.001 (two-tailed Student's $t$ test). **(E and G)** Localization of Mup1 relative to Erg6 in wild-type and *vps13Δ* cells. Cells expressing Mup1-GFP and Erg6-mKate or Sec63-mCherry were grown in the absence of methionine (0 min) in a minimal medium to an $OD_{600}$ of 0.8. Methionine was added, and cells were analyzed by fluorescence microscopy after 10- or 30-min. Individual slices are shown. Scale bar, 5 µm. **(F and H)** Mup1 puncta that partially overlap with Erg6 dots or Sec63. Mup1 dots ($n \geq 200$) were quantified by Image J as described in Fig. 1 C and Fig. 3 H. Only Mup1 puncta were considered for quantification. Bar graphs represent averages from three independent experiments. Error bars represent standard deviation (SD). ns, > 0.05; ***, P < 0.001 (two-tailed Student's $t$ test). **(I and J)** Localization of Cps1 relative to LDs in wild-type, *vps4Δ* and *vps4Δvps13Δ* cells. Cells expressing mCherry-tagged Cps1 were grown in a synthetic medium, and LDs were stained with Bodipy 493/503 (green). Cells were analyzed by fluorescence microscopy, and individual slides are shown. Scale bar, 5 µm. **(K)** Percentage of Cps1 partially overlapping with Bodipy dots or Sec63. Cps1 dots ($n \geq 150$) and Bodipy dots ($n \geq 800$) were quantified by Image J as described in Fig. 1 C and Fig. 3 H. Bar graphs represent averages from three independent experiments. Error bars represent standard deviation (SD). **, P < 0.01; ***, P < 0.001 (two-tailed Student's $t$ test).

Erg6-positive dots were observed next to small Mup1-GFP puncta after the addition of methionine for 10 min. This close proximity persisted but decreased after 30 min (Fig. 4, G and H; and Fig. 6, E and G). In contrast, the close proximity between Erg6 and Mup1 was significantly reduced in *vps13Δ* cells (Fig. 6, E and G). This picture changed when we analyzed MCSs between the ER and endosomes. In wild-type cells, Sec63-mCherry partially overlapped with Mup1-GFP 10 min after the addition of methionine but was overall more, and not less pronounced in *vps13Δ* cells (Fig. 6, F and G). Taken together, these data indicate that Vps13 is critical for LD–endosome crosstalk, while the formation of ER-endosome contact sites may still require additional tether protein(s) beyond Vps13.

Our TEM analysis showed that LDs established MCSs with Class E compartment (Fig. 4, C–F; and Fig. S3, A and F). To determine whether Vps13 indeed plays a crucial role in the potential LD–endosome contact sites, we examined the sorting of mCherry-Cps1 relative to LDs (stained by Bodipy) or the ER (marked with Sec63-GFP) in wild-type, *vps4Δ*, and *vps4Δvps13Δ* cells. Notably, deletion of *VPS4* resulted in Bodipy-positive LDs that were in part close to Class E compartments marked by mCherry-Cps1, in agreement with MCSs of LDs with intermediate endosomal structures. In agreement with findings on the Mup1 sorting assay (Fig. 6, E–G), this close contact was clearly reduced in *vps4Δvps13Δ* cells, whereas Sec63-labeled ER showed significantly more contacts with Class E compartments (Fig. 6, H–J). These data support our model that Vps13 facilitates crosstalk between LDs and endosomes, while additional tether protein(s) might be required to stabilize ER–endosome contact sites.

### Any1 and Vps13 are required for appropriate MVB formation

ER–endosome contact sites have been suggested to play key roles in cholesterol transfer, endosome positioning, receptor dephosphorylation, and endosome fission in mammalian cells (Raiborg et al., 2015), and Vps13-mediated endosome biogenesis in yeast (Suzuki et al., 2024). Given that Vps13 is also involved in a potential LD–endosome contact site, we wondered whether these MCSs have a similar function in promoting endosome formation. To investigate this, we analyzed the ultrastructure of endosomes by EM. In agreement with previous findings (Suzuki et al., 2024), *vps13Δ* cells had more MVBs compared with wild-type cells (Fig. 7 C), whereas the number of MVBs was much less in *any1Δ* cells. Consistent with this, the number of Ypt7-positive LE significantly decreased, with a similar observation in Any1$^{T60L,S64L}$ mutant cells (Fig. 3, D and E). However, the number of Vps4-positive endosomes remained unchanged in *any1Δ* cells (Fig. 7, E and F). As observed before (Suzuki et al., 2024), deletion of Vps13 resulted in larger MVBs that were approximately twice the diameter of normal MVBs (Fig. 7, A and D). Similarly, MVBs in *LDΔ* cells were slightly larger than in wild-type cells (Fig. 7, A and D). Finally, we asked if the deletion of LD-forming proteins, Any1 or Vps13 affected the morphology of the Class E compartment of *vps4Δ* cells (Fig. 7 B). In all three cases, cells were unable to form Class E compartments and accumulated enlarged compartments next to the vacuole, in agreement with our observation on Cps1 localization (Fig. 5, D, F, and H). Together, these data show that Any1 and Vps13 are required for appropriate MVB formation, likely at a novel LD–endosome contact site, as well as at ER–endosome contact sites.

## Discussion

ER–endosome MCSs play an essential role in endosome biogenesis by providing lipids via the Vps13 lipid transport protein (Suzuki et al., 2024). However, protein-mediated lipid transport in organelle expansion predicts that lipid transport must coordinate with a scramblase to rebalance lipids between the membrane leaflets of the lipid donor and acceptor (Ghanbarpour et al., 2021), though this scramblase was missing on endosomes. Here, we identified the "PQ-loop family" member Any1 as a novel phospholipid scramblase on the endosomes and revealed its critical role in endosome biogenesis. We show that Any1 cycles between the Golgi and endosomes with the help of the endosomal Retromer complex. If Any1 is lacking, endosomal trafficking is impaired, suggesting that Any1 is required for the endocytic pathway. Moreover, we identified mutants that abolished scrambling activity in silico and led to the accumulation of EE and a reduced number of MVBs. We further uncover that LDs accumulate next to endosomes, suggesting that crosstalk between LDs and endosomes via a potential contact site supports MVB formation. The previously characterized lipid

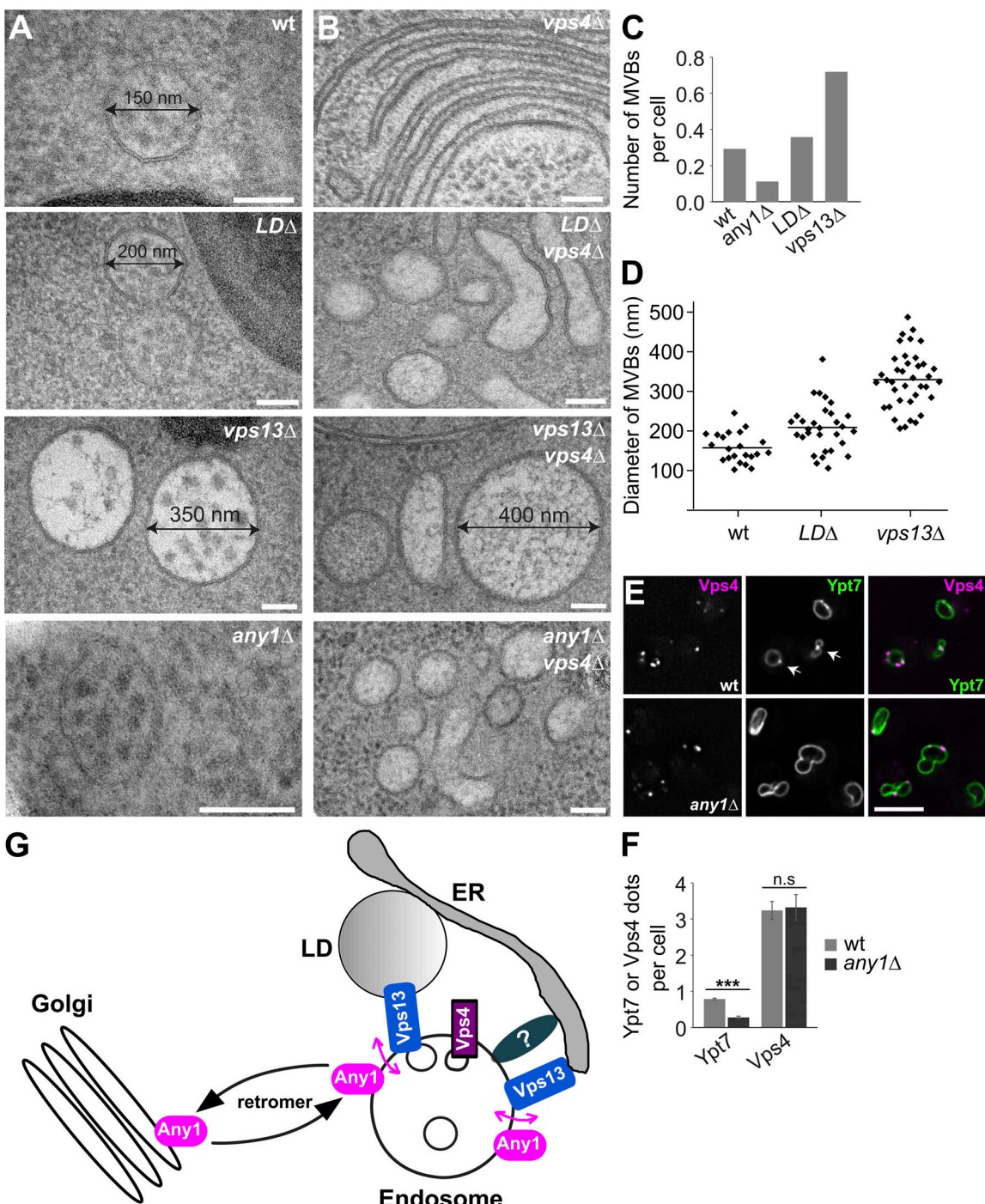

Figure 7. **LDs and Any1 are critical to the formation of MVBs. (A and B)** Analysis of endosomal structures in wild-type and mutant cells by TEM (see Materials and methods). Scale bars, 100 nm. **(C and D)** Quantification of the endosomal structures from A and B. **(C)** n = wt (118), LDΔ (115), any1Δ (114), and vps13Δ (97). **(D)** n = wt (22), LDΔ (40), vps13Δ (66). The diameter was always measured across the largest part of the MVBs. **(E)** Localization of Ypt7 relative to Vps4 in wild-type and any1Δ cells. Cells expressing GFP-tagged Ypt7 and mCherry-tagged Vps4 were grown in a synthetic medium and analyzed by fluorescence microscopy, and individual slices are shown. Scale bar, 5 μm. **(F)** Quantification of the number of Ypt7 and Vps4 from D. Ypt7 dots (n ≥ 100) and Vps4 dots (n ≥ 400) were quantified by Image J. The number of Ypt7 or Vps4 dots was manually counted from a summed projection. Error bars represent standard deviation (SD). Bar graphs represent averages from three independent experiments. ns, > 0.05; ***, P < 0.001 (two-tailed Student's t test). **(G)** Working model of Any1 function in endosome biogenesis. For details see text.

transfer protein Vps13 colocalizes with Any1 and may facilitate this process. We proposed that Vps13 function depends on the scramblase Any1 at endosomes to allow for lipid flux between LDs/ER and endosomes and promote endosome biogenesis.

Our data imply that Any1 cycles between the trans-Golgi network and endosomes require the Retromer complex. Retromer collects cargo on endosomes and forms tubules that transport signaling receptors, SNAREs, ion channels, and amino acid

transporters from endosomes to the PM or to the trans-Golgi network (Seaman et al., 1998; Seaman, 2004, 2012). Cycling of Any1 ensures that the scramblase is present both at the trans-Golgi and throughout the endolysosomal pathway except the vacuole, and thus escapes recognition and intraluminal sorting by the endosomal ESCRT machinery. Retromer recognizes its cargos by distinct signal sequences (Nothwehr et al., 1993; Cooper and Stevens, 1996; Redding et al., 1996; Finnigan et al., 2012; Suzuki et al., 2019). How Any1 is specifically recognized by the Retromer complex remains to be further dissected.

Any1 has been speculatively suggested as a phospholipid scramblase at the Golgi, though this activity was never demonstrated (Yamamoto et al., 2017). Here, we show that Any1 has scramblase activity using well-established in vitro and in silico approaches (Chumpen Ramirez et al., 2023; Li et al., 2024a). As our in vitro assay relies on NBD-PE as a fluorescent substrate, whether Any1 scrambles lipids non-specifically like ATG9A (Maeda et al., 2020) or exhibits substrate specificity would require further investigations. Although we identified polar residues in Any1 that impaired its function, it remains to be determined whether this polar region is required for scramblase activity in vitro. Similar to Atg9, which cooperates with the lipid transfer protein Atg2 at an MCS between ER and phagophore (Ugur et al., 2020; Dziurdzik and Conibear, 2021; Melia and Reinisch, 2022), Any1 and the Atg2-like Vps13 function together in the biogenesis of endosomes and MVB formation. As to whether Any1 directly interacts with Vps13 at the proposed LD–endosome contact site is regulated in its activity by phosphorylation or other modifications or is coordinated with ESCRTs remains to be investigated in future studies.

Vps13 is proposed to transfer lipids from the ER to endosomes at ER–endosome contact sites, which is required for ESCRT-mediated sorting during MVB biogenesis (Suzuki et al., 2024). However, deletion of Vps13 leads to the formation of much larger and more MVBs (Suzuki et al., 2024). Similarly, we also observed slightly larger and more MVBs in LD-deficient cells. Strikingly, in vps13Δ cells, the close proximity between endosomes and LDs is significantly reduced, whereas MCSs between endosomes and the ER are increased. How do we explain this observation? We propose that the lipid composition of endosomes in vps13Δ cells is altered due to an imbalance in lipid fluxes between LD–endosome contact sites and ER–endosome contact sites. This could in turn partially impair the fusion of MVBs with the vacuoles, but still allow for endosome-endosome fusion, which consequently becomes larger. In the absence of Any1, lipids from the MCSs cannot properly equilibrate at the endosomal membrane, leading to a delay in MVB formation and a significant reduction in the overall number of MVBs. Interestingly, in any1Δ cells, the number of Ypt7-positive LE was significantly decreased, while there was no change observed in Vps4-positive endosomes. We previously demonstrated that the subpopulation of Ypt7-positive endosomes relies on ESCRTs for their formation but lacks Vps4, indicating that they represent mature MVBs ready for fusion with the vacuole (Füllbrunn et al., 2024). Based on this, we proposed that Any1 supports ILV formation and, consequently, maturation toward Ypt7-positive MVBs, which can then fuse with the vacuole.

Our study identifies a potential novel contact site between LDs and endosomes, which exists in addition to the known ER–endosome MCS (Suzuki et al., 2024). Given that LDs are surrounded by a phospholipid monolayer (Pol et al., 2014; Walther et al., 2017), a lipid flow in either direction should include the regulated activity of lipid remodeling enzymes on LDs, such as phospholipases or hydrolases, similarly to autophagosome biogenesis (Shpilka et al., 2015; Deretic, 2015). In addition, it is possible that the ER is in proximity to the proposed LD-endosome contact site, thus generating a three-way contact site, where lipids could be equilibrated between endosomes and ER via LDs. How LDs precisely function to promote endosome biogenesis and how lipid flow via Vps13 and Any1 is regulated at this contact site will be an exciting direction in future studies.

## Materials and methods

### Yeast strains and molecular biology
Strains and plasmids used in this study are listed in Supplementary file 1 and 2, respectively. Deletion and tagging of genes in the cells were done by PCR-based homologous recombination with corresponding primers and templates (Puig et al., 1998; Janke et al., 2004). Vps13^mGFP and LDΔ mutant cells were kindly provided by Arun T. John Peter (University of Fribourg, Fribourg, Switzerland) and Florian Fröhlich (University of Osnabrück, Osnabrück, Germany). Plasmid encoding temperature-sensitive Vps4 was kindly provided by Markus Babst (University of Utah, Salt Lake City, UT, USA).

### Fluorescence microscopy
Yeast cells were grown to log phase in a synthetic complete medium (yeast nitrogen base without amino acids and with ammonium sulfate) containing 2% glucose (SDC+all) at 30°C. Selective temperature-sensitive (ts) strains were cultured in SDC+all medium at 23°C until the log phase and then shifted to 37°C for the indicated time. Cells were imaged at room temperature on a DeltaVision Elite imaging system based on an Olympus IX-71 inverted microscope with 100× NA 1.49 objectives, sCMOS camera (PCO), and Insight SSI (TM) illumination system, 4′,6-diamidino-2-phenylindole, GFP, mCherry, and Cy5 filters. Stacks of 6–10 images were taken at 0.2–0.35 μm spacing and deconvolution of the images was performed using the SoftWoRx software (Applied Precision).

### Protein purification
Any1 was overexpressed in yeast with a 3xFLAG tag under the control of the GAL1 promoter. Cells were grown to an $OD_{600}$ of 3 at 30°C in 2% galactose (4987.3; Carl Roth GmbH). Cells were then harvested and resuspended in lysis buffer (300 mM NaCl, 1.5 mM $MgCl_2$, 50 mM HEPES, pH 7.5), containing the protease inhibitor mix FY (39104.02; SERVA Serving Scientific) and 1 mM PMSF (63672; Carl Roth GmbH). Resuspended cells were frozen into small droplets in liquid nitrogen and lysed in a Freezer Mill 6875D (SPEX samplePrep) for 10 cycles (2 min each) at 12 counts per second (cps). The lysates were clarified by centrifugation at 3,200 $g$ for 15 min at 4°C, followed by centrifugation at 100,000 $g$ for 90 min to pellet membrane debris. Membranes

were resuspended in lysis buffer containing 2% n-dodecyl β-D-maltoside (CN26.4, DDM; Carl Roth GmbH) and incubated for 2 h at 4°C with gentle agitation. The resuspended membranes were centrifuged at 100,000 $g$ for 45 min to remove insoluble material. The supernatant was diluted to a final concentration of 0.2% DDM and incubated with anti-FLAG M2 affinity gel (A2220; Sigma-Aldrich) according to the manufacturer's instructions. Protein elution was performed in a lysis buffer containing 0.1% DDM and 0.3 mg/ml FLAG peptide.

## Coarse-grained molecular dynamics simulations (CG-MD)
The initial structural models for the monomer and homo-oligomer (dimer and trimer) of ScAny1 (Uniprot ID: Q03687) and HsSLC66A2 (Uniprot ID: Q8N2U9) were generated with AlphaFold2 (Evans et al., 2021, Preprint) via ColabFold (Mirdita et al., 2022). The models with the best confidence score (predicted local-distance difference test, pLDDT) are shown in supplementary information in Fig. S2 D. Any1 mutations (selected amino acid to Leucine) were generated using UCSF Chimera v1.16 software (Pettersen et al., 2004) and the Dunbrack 2010 rotamer library (Shapovalov and Dunbrack, 2011).

MD simulations were carried out with the software Gromacs 2021; Abraham et al., 2015) at CG resolution level with the Martini 3 force field (Souza et al., 2021). The all-atom structure of the best model per each protein/multimer was mapped to the martini CG resolution using the Martinize2 script (Kroon et al., 2024). To conserve the tertiary structure of the protein, an elastic network was applied using a force constant of 500 kJ mol$^{-1}$ nm$^{-2}$. The CG proteins were embedded into a DOPC membrane using the Insane script (Wassenaar et al., 2015). The charges of the systems were neutralized with Na$^+$ and Cl$^-$ martini beads and finally solvated with a solution of 0.15 M NaCl. All systems were first subjected to one energy minimization stage. Next, seven equilibration stages where positional restraints on the protein backbone and phosphate beads were gradually released were carried out. Finally, a production of 20 μs was carried out using a time step of 20 fs. A temperature of 310 K was controlled with the V-rescale thermostat (Bussi et al., 2007) and a coupling constant of 1 ps. A semi-isotropic Parrinello–Rahman barostat was used to keep the pressure constant at a value of 1 bar (Parrinello and Rahman, 1981) with a coupling constant of 12 ps. Two replicas with different initial velocities were carried out per each system. Electrostatic interactions were computed using the reaction-field approach with 1.1 nm as the cut-off. Van der Waals interactions were computed using the cut-off scheme with a cut-off distance of 1.1 nm. The Potential-shift-Verlet was used as Verlet cut-off scheme for both types of interactions.

## Analyses of CG-MD
### In silico lipid scrambling assay
To compute scrambling activity from our CG-MD simulations, we traced the angle between each lipid and the z-axis of the membrane along the entire simulation, similarly as it was previously described (Li et al., 2024a). To not overestimate the number of events, a buffer or tolerance region was defined between 50° and 130°. Angles were computed every 1 ns with the gmx_gangle tool. Block averaging was carried out with samples of 1 μs. Thus, for a simulation of 20 μs a total of 20 points were computed and then represented through boxplots (Fig. 2 C).

### Main amino acids involved in lipid scrambling
To identify which amino acids were involved in each individual scrambling event, a protein–lipid contact analysis was carried out with the gmx_select tool. For this, a contact was considered when either the headgroup or phosphate bead from the scrambled lipid was at a distance of 0.45 nm or lower from any protein bead. The total number of contacts was saved in the beta-factor column of the PDB format and used for the representation of Fig. 2 D. The in-house bash script used both for the lipid scrambling and main amino acids involved in lipid scrambling analyses can be found in Zenodo, https://doi.org/10.5281/zenodo.10475371, Li et al., 2024b.

### Membrane curvature
Membrane curvature was computed with the g_lomepro software (Gapsys et al., 2013). Prior to curvature membrane analysis, the rotation and translation of the protein were fitted using the gmx_trjconv tool. During curvature analysis, the bin size (x and y) was set up at 120 and frames every 1 ns were used.

For all analyses, the first 2 μs from each replica were omitted for equilibration.

## Lipid scramblase assay
Liposomes for lipid scramblase assays were prepared as described (Ploier and Menon, 2016; Chumpen Ramirez et al., 2023). Palmitoyl-oleoyl phosphatidylcholine (POPC), palmitoyl oleoyl phosphatidylglycerol (POPG) (850457, 840457; Avanti Polar lipids) and the fluorescently-labeled lipid N-(7-nitrobenz-2-oxa-1,3-diazol-4-yl)-1,2-dihexadecanoyl-sn-glycero-3-phosphoethanolamine (N360, NBD-PE; Invitrogen) were mixed in the ratio of 8.95:1:0.05, dried under vacuum, resuspended in buffer A (50 mM HEPES/NaOH, pH 7.4, 100 mM NaCl), and sonicated for 10 min. Liposomes were extruded through a membrane with a pore size of 400 nm (25 passages) (10417104; Whatman) and then through a membrane with a pore size of 200 nm (20 passages) (800281; Whatman). To incorporate Any1 into liposome membranes, liposomes were first incubated with 0.35% DDM for 3 h at room temperature to destabilize liposomes. Then, purified Any1 protein was added and incubated with liposomes for 1 h at room temperature. For protein-free liposome controls, destabilized liposomes were incubated with an equivalent volume of the elution buffer used to purify Any1. The detergent was gradually removed by the addition of BIO-BEADS SM2 adsorbent Media (1523920; Bio-Rad) as described (Ploier and Menon, 2016).

For the 2-NBD glucose assay, the liposomes and proteoliposomes were prepared as described for the lipid scrambling assay, except that 1 mM 2-(N-(7-nitrobenz-2-oxa-1,3-diazol-4-yl)amino)-2-deoxyglucose (2-NBDG; N13195; Invitrogen) was added to the dried lipid mix with buffer A. The following steps were the same as for the preparation of NBD-PE liposomes.

For the scramblase assay, the liposome suspensions were diluted 20 times in buffer A and then stirred in a cuvette at 23°C.

The fluorescence of NBD was recorded over time using a Spectrofluorometer FP-6500 (Jasco, Inc.) with an excitation wavelength of 470 nm and an emission wavelength of 530 nm. Fluorescence intensity was normalized to the average of the initial intensity prior to the addition of dithionite (first 200 s). 200 s later, a final concentration of 30 mM sodium dithionite was added and recorded for 400 s, followed by the addition of 0.1% Triton X-100 and recorded for an additional 200 s.

### Electron microscopy

Yeast cells were grown in YPD (1% yeast extract, 2% peptone, 2% glucose) medium to $OD_{600}$ of 0.4–0.6. Cells were concentrated by vacuum filtration onto a 0.45-μm membrane filter (HVWG04700; Merck). After filtration, the membrane filter was placed onto an agar plate. The yeast paste was scraped together using a pipette tip and 3 μl was transferred into a 100-μm deep cavity of an aluminum planchette (Engineering Office M. Wohlwend GmbH, 241) until the cavity was overfilled. Afterward, the flat side of a planchette (Engineering Office M. Wohlwend GmbH, 242) was placed on top and excess yeast paste was quickly removed. The finished assembly was immediately subjected to high-pressure freezing using an HPF Compact 03 (Engineering Office M. Wohlwend GmbH). Vitrified samples were stored in liquid nitrogen until further processing.

For freeze substitution (FS), planchettes containing vitrified yeast paste were separated in liquid nitrogen under a stereo microscope. An AFS2 instrument (Leica) was used for substitution. The chamber was filled with ethanol and precooled to –90°C. FS solution consisted of 1% osmium tetroxide (E19134; Science Services), 0.1% uranyl acetate (E22400; Science Services), 5% water in anhydrous acetone (83683.230; VWR), and was prepared on ice. A 20% uranyl acetate stock in anhydrous methanol (83679.230; VWR) was used. FS solution was added to 1.5 ml cryovials (72.694.600; Sarstedt), precooled to –90°C, and opened planchettes were added. Freeze substitution was conducted for 24 h at –90°C, 12 h at –60°C, and 12 h at –30°C, finally reaching 0°C. Samples were further incubated for 1 h at RT to increase contrast. Afterwards, samples were washed five times in anhydrous acetone and stepwise-infiltrated with EPON812 (Roth) in 1:3, 1:1, and 3:1 dilutions with anhydrous acetone, each lasting at least 8 h at room temperature. Samples were incubated in fresh EPON overnight and finally polymerized at 60°C for 48 h. The polymerized EPON blocks were trimmed to a trapezoid shape and 70 nm sections were prepared using an ultramicrotome (ULTRACUT EM UC7; Leica) and transferred to formvar-coated slot grids.

For TEM, resin sections on EM grids were contrasted with 3% uranyl acetate for 30 min and 2% lead citrate for 20 min in the LEICA EM AC20. Subsequently, sections were analyzed using a JEOL TEM JEM2100-Plus at 200 keV equipped with a 20-megapixel EMSIS Xarosa CMOS camera (EMSIS) or a Zeiss Leo Omega AB equipped with a CCD camera (Tröndle). For quantification of abundance and distance measurements, a fitting number of single images was collected by hand from one section, and the abundance of endocytic compartments was counted or their distance to cellular organelles was measured manually using the RADIUS software (EMSIS).

### Statistical analysis

Statistical analyses of all fluorescent microscopy images were performed with Excel (Microsoft). The assumption of normal data distribution was made, although it was not formally tested. To assess differences between the two groups, we employed a two-sided Student's $t$ test. For comparisons involving multiple groups, a one-way ANOVA was conducted, followed by Tukey's post-hoc test. Statistical significance is indicated as follows: *$P < 0.05$, **$P < 0.01$, and ***$P < 0.001$. All statistical analyses and corresponding P values are detailed in the figure legends.

### Online supplemental material

Fig. S1 shows additional data for Fig. 1. Fig. S2 shows additional data for Fig. S2. Figs. S3 and S4 show additional data for Fig. 4. Table S1 includes strains and Table S2 shows plasmids used in this study.

### Data availability

All data used in this study are available upon request.

## Acknowledgments

We thank Ayelén González Montoro, Lars Langemeyer, and Florian Fröhlich for comments and support, Sabrina Chumpen for help with the scramblase assay, Kathrin Auffarth and Angela Perz for expert technical support, and the entire Ungermann lab for feedback.

This work was supported by the Deutsche Forschungsgemeinschaft in SFB 1557 (Project P14 to C. Ungermann, P8 to M. Hensel, Z2 to O.E. Psathaki and by the Swiss National Science Foundation (grant CRSII5_189996 to S. Vanni), and the European Research Council under the European Union's Horizon 2020 Research and Innovation program (grant agreement no. 803952, to S. Vanni). This work was supported by a grant from the Swiss National Supercomputing Centre under the project ID s1269. Open Access funding was provided by the Universitätsbibliothek Osnabrück.

Author contributions: J. Gao: Conceptualization, Data curation, Formal analysis, Investigation, Methodology, Project administration, Supervision, Validation, Visualization, Writing - original draft, Writing - review & editing, R. Franzkoch: Data curation, Formal analysis, Investigation, Methodology, Visualization, Writing - original draft, Writing - review & editing, C. Rocha-Roa: Formal analysis, Software, Writing - review & editing, O.E. Psathaki: Resources, Writing - review & editing, M. Hensel: Methodology, Project administration, Resources, Supervision, Writing - review & editing, S. Vanni: Conceptualization, Funding acquisition, Methodology, Supervision, Validation, Writing - original draft, Writing - review & editing, C. Ungermann: Conceptualization, Funding acquisition, Project administration, Supervision, Validation, Visualization, Writing - original draft, Writing - review & editing.

Disclosures: The authors declare no competing interests exist.

Submitted: 2 October 2024

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

# Supplemental material

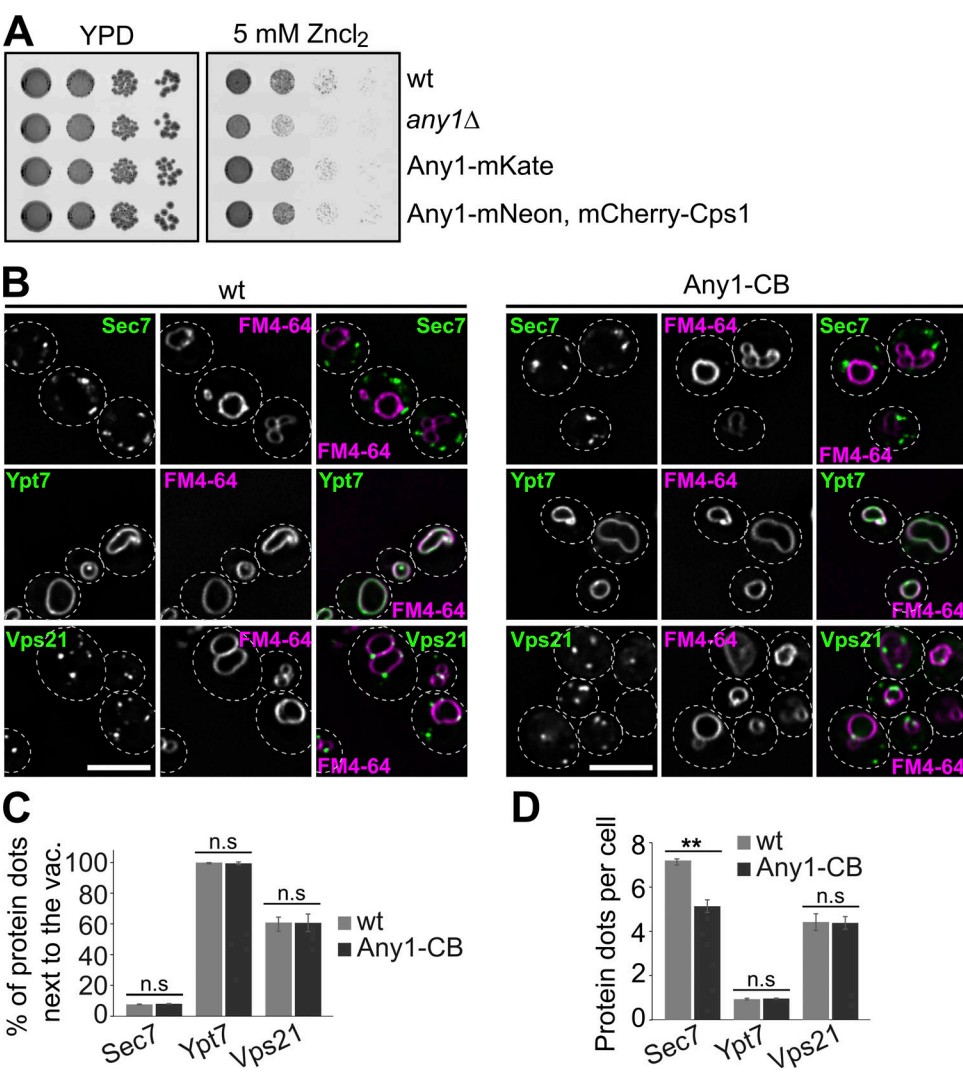

Figure S1. **Localization of Sec7, Ypt7, and Vps21 in cells expressing Any1-CB. (A)** Growth assay on ZnCl₂-containing plates. The indicated cells were grown in synthetic medium, spotted onto plates containing YPD with or without 5 mM Zn²⁺, and grown at 30°C for 2–3 days. **(B)** Localization of Sec7, Ypt7, and Vps21 relative to the vacuoles in wild-type and Any1-CB cells. Cells expressing GFP-tagged Sec7 or Vps21 or Ypt7 or with chromobody-tagged (CB) Any1 were grown in a synthetic medium and analyzed by fluorescence microscopy, and individual slices are shown. Scale bar, 5 μm. **(C and D)** Quantification of the localization and number of Sec7, Ypt7 and Vps21 from A. Sec7 dots (n ≥ 900), Vps21 dots (n ≥ 900), and Ypt7 dots (n ≥ 200) were quantified by Image J. The number of Sec7, Ypt7, and Vps21 dots was counted manually from a sum projection. Bar graphs represent the averages from three independent experiments. Error bars represent standard deviation (SD). ns, > 0.05; **, P < 0.01 (two-tailed Student's t test).

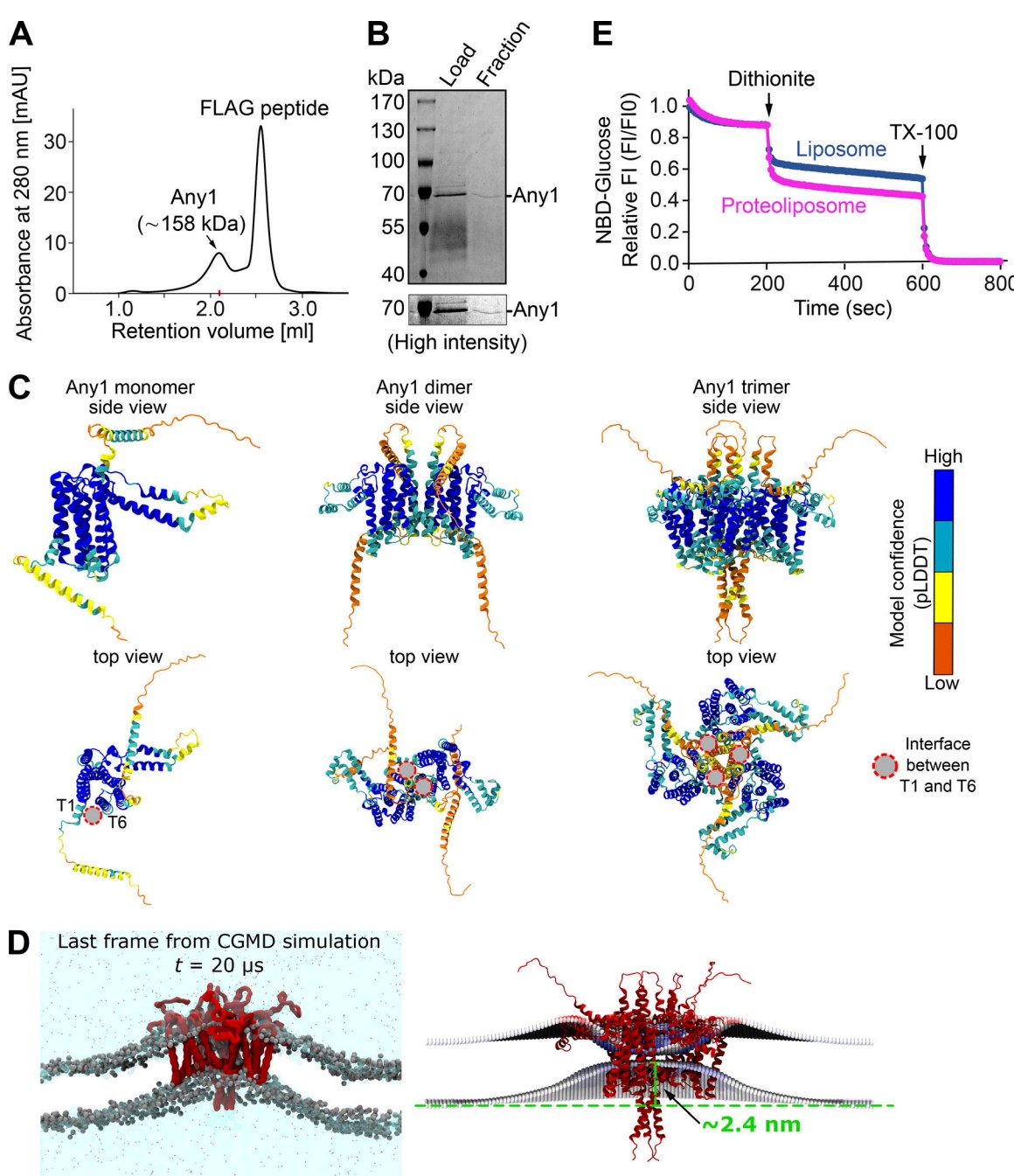

Figure S2. **Access of liposome enclosed 2-NBDG to dithionite. (A)** Size exclusion chromatography (SEC) of affinity-purified Any1. Purification was done as described in Materials and methods. **(B)** Purified Any1. Proteins were from affinity purification (eluate, left) and SEC (right). **(C)** Best structural models for the monomer, dimer, and trimer of ScAny1 predicted with AlphaFold2. The models are colored by their Predicted local-distance difference test (pLDDT) confidence score. **(D)** Membrane curvature induced by the trimeric structure of ScAny1. Right, representative screenshot of the last frame from one of the replicas for ScAny1 simulations. The protein is shown as a red surface, the phosphate beads are shown as gray spheres, and the NaCl solution is shown as a cyan shadow with the ions as dots. Left, membrane curvature analysis was performed with the g_lomepro tool. An estimated elevation in the lower membrane leaflet of ~2.5 nm was obtained. **(E)** 2-NBDG protection assay. The fluorescence of 2-NBDG captured in liposomes or proteoliposomes (protein: phospholipid ratio = 1:1,500) was measured over time, and the permeability of the membrane was assessed by the addition of 30 mM sodium dithionite, followed by the addition of 0.1% of Triton X-100. Source data are available for this figure: SourceData FS2.

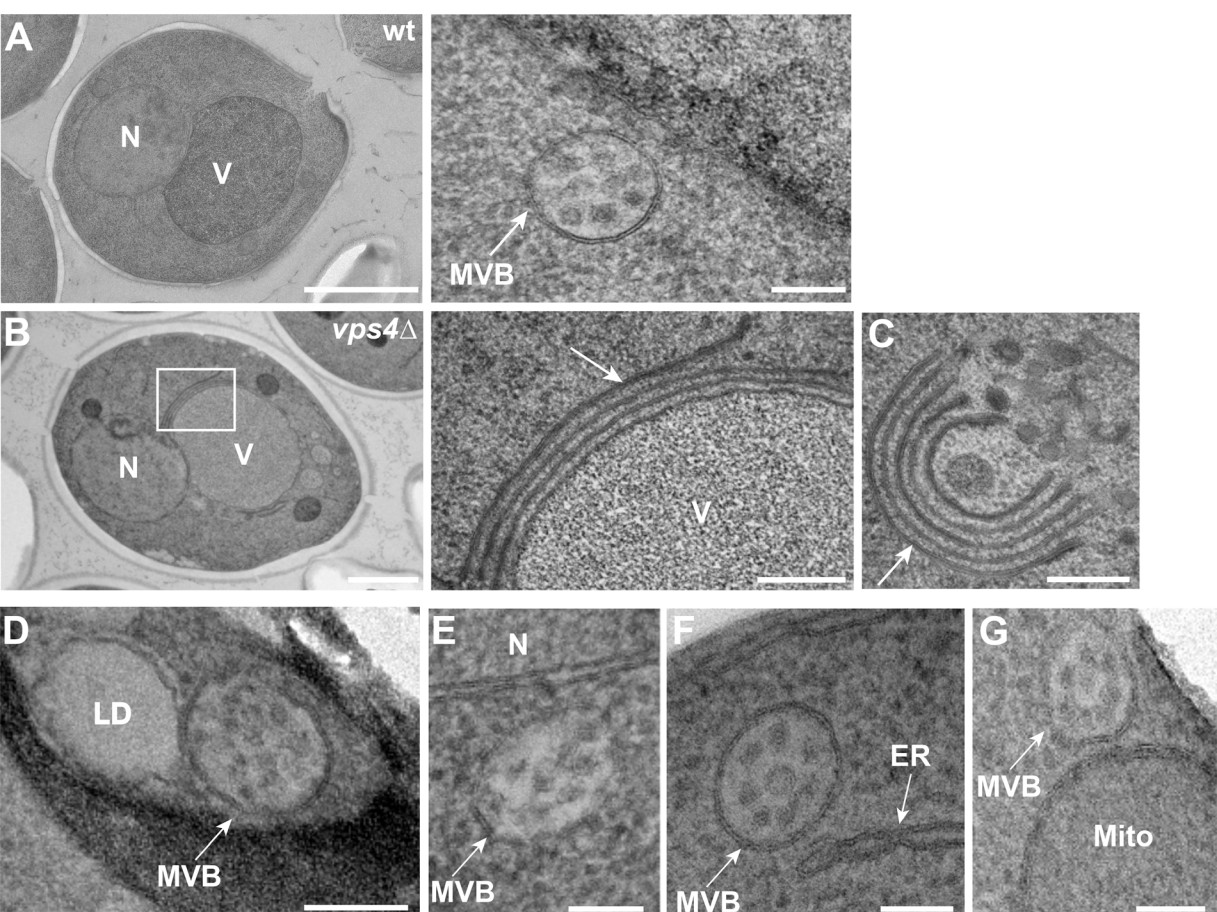

Figure S3. **Endosome proximity to selected organelles. (A–C)** Representative ultrastructure images of the endosomes in wild-type cells (A) and Class E compartments in *vps4Δ* cells. **(B)** Scale bar, 1 µm. **(C)** Scale bar, 100 nm. **(D–G)** Examples of MCSs between endosomes and LD or nucleus or ER or mitochondria by TEM. Scale bar, 100 nm.

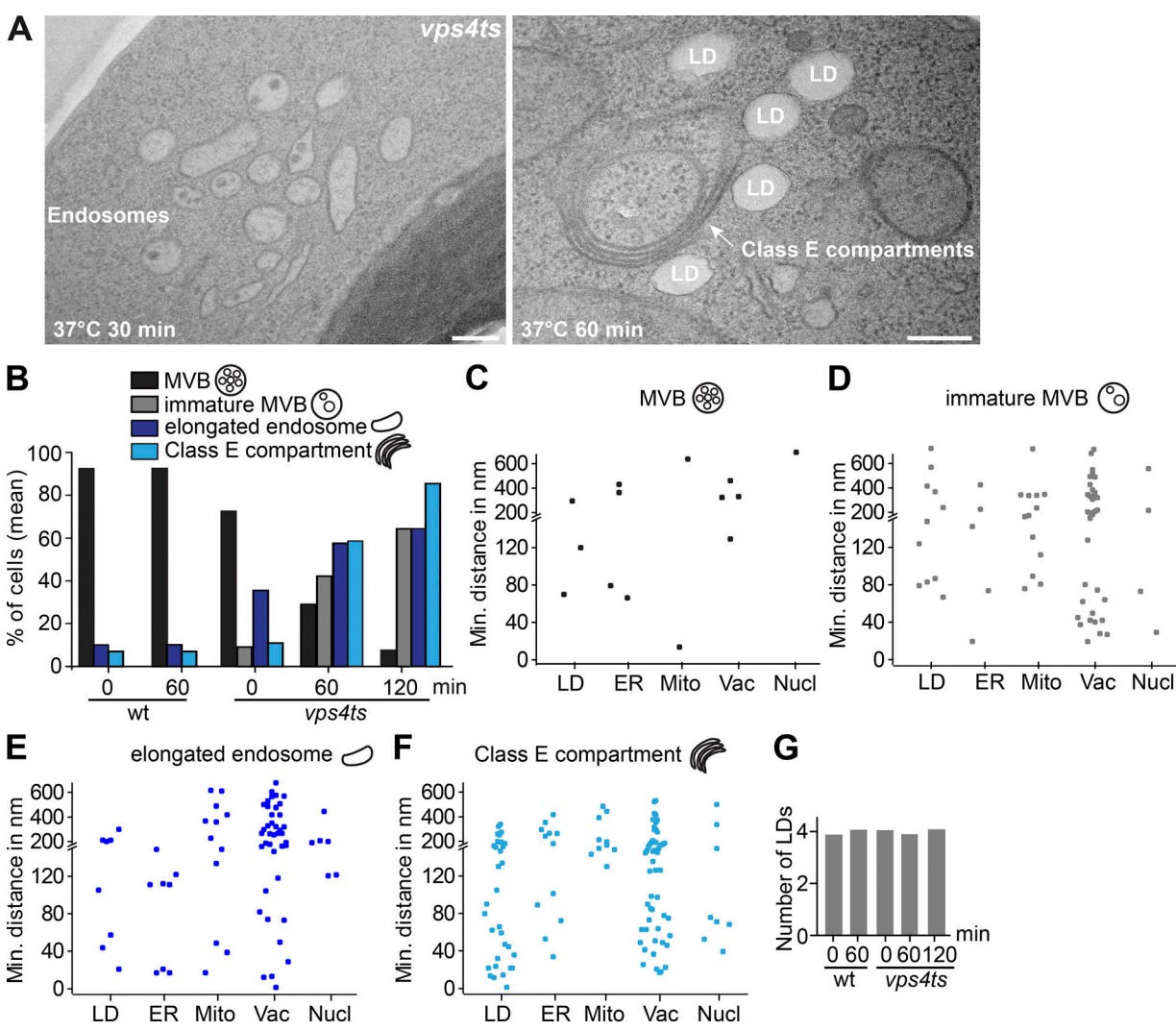

Figure S4.  **LDs localization relative to the endosomal class E compartment. (A)** Analysis of the ultrastructure of the endosomes (left), and MCSs between LDs and class E compartments (right) in *vps4 ts* mutant cells. *vps4 ts* cells were grown at 23°C in a synthetic medium, and then shifted to 37°C for 30 or 60 min. Scale bar, 200 nm. **(B and G)** Percentage of cells containing different endosomal structures (B) or quantification of number of LDs (G) in wild-type and *vps4* ts mutant cells. Cells were grown in synthetic medium at 23°C and then shifted to or not shifted to 37°C for 30 or 60 min, or *vps4* ts mutant cells were shifted to 37°C for 120 min. wt, 0 min: number of cells (*n*) = 96; 60 min: *n* = 60; *vps4 ts*, 0 min: *n* = 111; 60 min: *n* = 81; 120 min: *n* = 87 were quantified. **(C–F)** Quantification of the distance between the intermediate endosomal structures and LDs or ER or mitochondria or vacuoles or nuclei in *vps4 ts* cells at 37°C for 2 h from A. A total of 87 cells were quantified.

Provided online are Table S1 and Table S2. Table S1 shows yeast strains used in this study (Gao et al.). Table S2 shows plasmids used in this study (Gao et al.).

