## [Peer Review File · The Journal of Cell Biology]

Any1 is a phospholipid scramblase involved in endosome biogenesis

Jieqiong Gao, Rico Franzkoch, Cristian Rocha-Roa, Olympia Psathaki, Michael Hensel, Stefano Vanni, and Christian Ungermann

Corresponding Author(s): Christian Ungermann, Osnabrück University

Review Timeline:

Submission Date:	2024-10-02
Editorial Decision:	2024-11-12
Revision Received:	2024-12-17
Editorial Decision:	2025-01-16
Revision Received:	2025-01-22

Monitoring Editor: Harald Stenmark

Scientific Editor: Andrea Marat

Transaction Report:

DOI: <https://doi.org/10.1083/jcb.202410013>

November 12, 2024

Re: JCB manuscript #202410013

Christian Ungermann
Osnabrück University

Dear Prof. Ungermann,

Thank you for submitting your manuscript entitled "Any1 is a novel phospholipid scramblase involved in endosome biogenesis". The manuscript was assessed by expert reviewers, whose comments are appended to this letter. We invite you to submit a revision if you can address the reviewers' key concerns, as outlined here.

You will see that the reviewers are overall positive that the identification of Any1 as a scramblase for endosomal lipid transport is an interesting and important finding. They have provided constructive suggestions, which we hope you agree will further improve your study. While the relationship between Any1 and Vps13 is an interesting question, we do not require that it be addressed in the current study. Otherwise, please address the remaining reviewer comments in a revised manuscript as best you can.

GENERAL GUIDELINES:

Text limits: Character count for an Article is < 40,000, not including spaces. Count includes title page, abstract, introduction, results, discussion, and acknowledgments. Count does not include materials and methods, figure legends, references, tables, or supplemental legends.

Figures: Articles may have up to 10 main text figures. Figures must be prepared according to the policies outlined in our Instructions to Authors, under Data Presentation, <https://jcb.rupress.org/site/misc/ifora.xhtml>. All figures in accepted manuscripts will be screened prior to publication.

*****IMPORTANT:** It is JCB policy that if requested, original data images must be made available. Failure to provide original images upon request will result in unavoidable delays in publication. Please ensure that you have access to all original microscopy and blot data images before submitting your revision. ***

Supplemental information: There are strict limits on the allowable amount of supplemental data. Articles may have up to 5 supplemental figures. Up to 10 supplemental videos or flash animations are allowed. A summary of all supplemental material should appear at the end of the Materials and methods section.

Please note that JCB now requires authors to submit Source Data used to generate figures containing gels and Western blots with all revised manuscripts. This Source Data consists of fully uncropped and unprocessed images for each gel/blot displayed in the main and supplemental figures. Since your paper includes cropped gel and/or blot images, please be sure to provide one Source Data file for each figure that contains gels and/or blots along with your revised manuscript files. File names for Source Data figures should be alphanumeric without any spaces or special characters (i.e., SourceDataF#, where F# refers to the associated main figure number or SourceDataFS# for those associated with Supplementary figures). The lanes of the gels/blots should be labeled as they are in the associated figure, the place where cropping was applied should be marked (with a box), and molecular weight/size standards should be labeled wherever possible.

The typical timeframe for revisions is three to four months. If you anticipate any difficulties in meeting this aforementioned revision time limit, please contact us and we can work with you to find an appropriate time frame for resubmission. Please note that papers are generally considered through only one revision cycle, so any revised manuscript will likely be either accepted or rejected.

Thank you for this interesting contribution to Journal of Cell Biology. You can contact us at the journal office with any questions at cellbio@rockefeller.edu.

Sincerely,

Harald Stenmark, PhD
Monitoring Editor

Andrea L. Marat, PhD
Deputy Editor

Journal of Cell Biology

Reviewer #1 (Comments to the Authors (Required)):

Recent work has implicated protein-mediated lipid transport in the formation of ILVs on MVBs as well as in the repair or maintenance of other organelles in the endo-lysosomal system. An expectation, from work in autophagy especially, is that these transporters will likely work in concert with lipid scramblases to effect membrane expansion. Here the authors identify Any1 as an endo-lysosomal pathway specific scramblase. They use a clever organelle targeting strategy involving GFP-binding motifs to trap Any1 at either the Golgi or within the endosomal pathway, confirming its biological relevance to the latter. In silico and in vitro work establishes that this protein can scramble lipids. These conclusions are interesting and strongly supported by their data.

Next, they work to connect Any1 function to a lipid transport process. They conclude that Any1 functions at the same stage as the bridge like lipid transporter Vps13 and further that this functionality depends in part on lipid droplet biogenesis. They suggest it is likely that this is due to Vps13 sitting at lipid droplet/endosome contact sites. Overall, this work makes a significant contribution to the burgeoning field of bridge-like lipid transport in the endo-lysosomal system and will be of fairly broad interest.

If there is a significant omission in the manuscript, it is the lack of EM evidence for this putative LD-endosome contact site. They show very nice EM of various contact sites in WT and class E compartment cells, but do not observe LD-MVB contacts, suggesting they are very rare or absent. However, via fluorescence microscopy, as much as 20% of all Mup1 containing endosomes are adjacent to Erg1 structures (which they suggest are likely LDs) at an optimal time point (10 minutes). Critically, the authors believe Vps13 and Any1 might be coordinating at just such a contact site, so clear evidence of that site is needed. Can these potentially transient colocalizations of Mup1 and Erg1 be visualized in EM, perhaps via CLEM?

I found it difficult to follow the rationale for studying class E compartment localizations. These structures accumulate in ESCRT knockdowns in yeast. Do the authors contend that these are stalled intermediates in the formation of a MLVs and is this supported by other literature? Otherwise, it is difficult to see specifically how these inform on MLV biogenesis. In addition, conditions where proteins fail to target to the class E compartment based on fluorescence leave unanswered the questions of whether this is a targeting problem or whether the compartment is absent. EM could validate whether the class E compartment still forms in Vps4/LD depletion studies.

Finally, I very much appreciated the discussion on Vps13 and MVB size/number. The original discovery by Suzuki et al has been difficult to rationalize mechanistically. The authors here make the same observations, and show a similar phenotype in LD knockdown conditions, supporting their model of a role for Vps13-related LD lipid transport.

Minor technical questions:

The image analysis methods are not entirely clear.

- 1) "% co-localization" is defined as something done in imageJ. Can you provide details on what this metric is (e.g. Pearson's) and the directionality of the co-localization (i.e. X colocalizes with Y may be different from Y colocalizes with X).
- 2) In the case of Mup1, the graphs describe Mup1 dots in the colocalization, presumably because the PM distribution of Mup1 at time 0 would be essentially perfectly colocalized with cortical ER. Please describe how Mup1 dots are identified.

Minor edits:

Line 318 - wrong figure panel is called out

Line 459 - I'm not sure you can say "thus scramblase activity is important" without a scramblase specific mutant. Though I agree your data certainly implies a role for scrambling.

Reviewer #2 (Comments to the Authors (Required)):

Jieqiong Gao and co-workers show in their manuscript that 'Any1 is a novel phospholipid scramblase involved in endosome biogenesis'. They use a combination of genetic, cell biological, and biochemical experiments and MD simulation to conclude that Any1 functions together with Vps13 to bring lipid droplets close to endosomes, which function together to supply and scramble lipids for proper MVB biogenesis.

These findings are interesting and provide novel molecular and cell biological insight into the organization of the endosomal pathway, but additional experiments are required to support these conclusions:

Major points:

Figure 1: Are the tagged version of Any1 functional?

Figure 1: The conclusion that Any1 is a retromer cargo, based on the data presented, is an overstatement. Either tone down the conclusion, or demonstrate that Any1 is a direct retromer cargo.

Figure 1G: Does the Sec7-GFP-Any1-CB complex disrupt Sec7 function, and could that partially explain the more severe growth defect compared to the any1 mutant. Also is the number of Any1-CB complexes with GFP-Ypt7, Sec7-GFP and GFP-Vps21 comparable? Without this information, the results of this experiment are difficult to interpret.

Figure 1H: The microscopy of Mup1-GFP needs to be complemented either with FACS analysis of Mup1-pHluorin and/or WB analysis. Are other endocytic and biosynthetic MVB cargoes also affected by loss of Any1 function?

Figure 2: Can Any1 mutants be generated (in silico, in vitro and in vivo) that block lipid scrambling and hence cause a loss of function or a hypomorphic phenotype? These mutants could also be invaluable tools to strengthen the overall conclusion of the manuscript.

Figure 3B, F: Please indicate the threshold distance for possible MCS.

Figure 3: Based on the current state of the manuscript, the statement that 'LDs form MCS with endosomes' is an overstatement. If anything, there is a spatial correlation of LD with class E compartments. The fact that LD are in close proximity to Mup1-containing endosomes as reported by live cell epifluorescence microscopy means that they could be 200 nm apart from each other. This experiment is insufficient to demonstrate the formation of LD-endosome MCS.

Could you block early to late endosome maturation with different mutants and then examine if LD MCS increase.

Figure 4 A - H: please provide co-staining with BODIPY and also include the respective any1 single mutant.

Figure 5E: in the vps13 mutant, Mup1 is barely internalized, and this could be one reason why less colocalization with LDs is detected. Strictly speaking, this does not mean that Vps13 establishes LD-endosome MCS.

Figure 6: Based on nice high pressure freezing and TEM experiments, the authors conclude that Any1 and Vps13 are required for appropriate MVB formation. Yet, TEM data does not fully support this conclusion, since measuring MVB diameter on thin sections is difficult to interpret.

Also do any1 mutants really have defects in endosome biogenesis? The MVBs look rather normal (but seem to be less frequent) by electron microscopy. Yet Mup1 preferentially accumulates in endosomes in the any1 mutants. Doesn't that point to a maturation defect along the endosomal compartment?

Finally, is there a direct functional relationship between Any1 and Vps13? Do they form a complex on endosomes? Is it clear that any1 functions downstream of Vps13, or could it also be the other way around?

Minor points

Figure 1A - Vps4 instead of 4

Figure 1E - any

Please explain how the colocalization was quantified with image J (Pearson correlation ?)

In Figure 1D, does expression of Any1-mNeonGreen- cause an MVB sorting defect? CPS1 appears to accumulate on the limiting vacuolar membrane?

Legend for Figure 1 does not contain sufficient information on image quantification (e.g. 1F)

The authors conclude: 'Surprisingly, the size of Any1 after SEC was around 158 kDa, while Any1 has a predicted molecular weight of 46.8 kDa (Figure S2 A, B), suggesting that Any1 might form a homo trimer.' Couldn't it be equally well micelle competition?

Figure S2D: the green labeling is difficult to see.

Figure 5A -> there is a typo in line 351 (figure 1 is called, instead of Figure 5)

Reviewer #3 (Comments to the Authors (Required)):

The authors describe the analysis of Any1, a protein that is required for proper MVB formation in yeast. Any1 traffics between Golgi and an endosomal compartment, in part by Retromer sorting, and co-localizes with an early but not with a late-endosome marker. In vitro and in silico studies supported the idea that Any1 functions as scramblase. Electron microscopy imaging suggested that ANY1 deletion affects endosome morphology and MVB numbers. Furthermore, electron microscopy images indicated the presence of contact sites between endosomes and lipid droplets. These contact sites dependent on the presence of Any1 and the lipid transfer protein Vps13. A model is provided that suggests that Any1 is a scramblase at the endosome that is required for proper endosomal maturation by aiding lipid exchange with ER and lipid droplets.

Overall, the data is compelling and supports the proposed role of Any1 in the lipid transport to and from the endosome. The Mup1 data fit to a simple model in which any1 mutants are delayed in the formation/maturation of MVBs because of lack of sufficient membrane for ILV formation. However, this model does not fit other observations. This simple model would predict an accumulation of immature MVBs and the accumulation of Cps1 in these pre-vacuolar compartments. In contrast, the data show that any1 mutants have less MVB-like structures and when combined with the vps4 mutant, any1 is able to suppress the formation of a class E compartment, which is indicated by the lack of accumulation of Cps1 and membranes in this double mutant. Furthermore, any1 shows less Ypt7 positive late-endosomes. Together, these observations fit better with the idea that loss of Any1 bypasses a checkpoint in endosomal maturation that would normally block immature MVBs from fusing with the vacuole. This checkpoint is the reason why in vps4 endosomal membranes accumulate and form the class E compartment, instead of just fusing with the vacuole. So, loss of Any1 might change lipid composition in a way that promotes fusion?

In summary, the phenotypes of the any1 mutant, the mutations in the lipid droplets formation and in Vps13 are complicated, a fact that should be better acknowledged in the discussions. At this point I feel the conclusions are too vague and the complexity of the data cannot be appreciated by researchers that are not intimately familiar with yeast endosomal trafficking.

Minor point: What is the localization of Cps1 in any1 mutant cells?

We thank the reviewers for their insightful comments, which have significantly improved our manuscript. In response to their critiques, we now:

- show that the tagged versions of Any1 are functional (Figure S1).
- complemented the Mup1-GFP sorting assay with an analysis of Mup1-pHluorin (Figure 1J and K).
- identified specific residues in Any1 based on its potential mechanism. These residues are required for scramblase activity *in silico* (Figure 3A) and *in vivo* function of Any1 in endosome maturation (Figure 3B-G).
- included the any1 deletion in the Cps1 sorting analysis, and compared lipid droplet localization in wild-type and corresponding mutants (Figure 5A-K).
- provide detailed information and quantification methods for image analysis in the figures and text.
- revised and expanded the text to clarify points raised by the reviewers.

Please find below a detailed response to all specific comments of the reviewers.

Reviewer#1:

Recent work has implicated protein-mediated lipid transport in the formation of ILVs on MVBs as well as in the repair or maintenance of other organelles in the endo-lysosomal system. An expectation, from work in autophagy especially, is that these transporters will likely work in concert with lipid scramblases to effect membrane expansion. Here the authors identify Any1 as an endo-lysosomal pathway specific scramblase. They use a clever organelle targeting strategy involving GFP-binding motifs to trap Any1 at either the Golgi or within the endosomal pathway, confirming its biological relevance to the latter. *In silico* and *in vitro* work establishes that this protein can scramble lipids. These conclusions are interesting and strongly supported by their data.

Next, they work to connect Any1 function to a lipid transport process. They conclude that Any1 functions at the same stage as the bridge like lipid transporter Vps13 and further that this functionality depends in part on lipid droplet biogenesis. They suggest it is likely that this is due to Vps13 sitting at lipid droplet/endosome contact sites. Overall, this work makes a significant contribution to the burgeoning field of bridge-like lipid transport in the endo-lysosomal system and will be of fairly broad interest.

Thank you for the overall positive evaluation of our work.

If there is a significant omission in the manuscript, it is the lack of EM evidence for this putative LD-endosome contact site. They show very nice EM of various contact sites in WT and class E compartment cells, but do not observe LD-MVB contacts, suggesting they are very rare or absent. However, via fluorescence microscopy, as much as 20% of all Mup1 containing endosomes are adjacent to Erg1 structures (which they suggest are likely LDs) at an optimal time point (10 minutes). Critically, the authors believe Vps13 and Any1 might be coordinating at just such a contact site, so clear evidence of that site is needed. Can these potentially transient colocalizations of Mup1 and Erg1 be visualized in EM, perhaps via CLEM?

We appreciate the reviewer's suggestion regarding the EM evidence for LD-endosome contact sites. While we agree that such evidence would be ideal, obtaining it is extremely challenging. Our current data indicate that LDs are in close proximity to Mup1-positive endosomes. We have therefore adjusted the text to reflect this conclusion, rather than referring to the existence of LD-endosome contact sites.

Additionally, our findings suggest that LDs interact with early endosomes, rather than mature MVBs, to facilitate endosome maturation, which could explain why LD-MVB contact sites are rarely observed. Although we have established powerful techniques to visualize yeast organelle ultrastructure using TEM, analyzing specific membrane contact sites with two-color colocalization under CLEM remains a significant challenge. This limitation is shared by others in the field, such as Fulvio Reggiori, who has long term experience with the ultrastructural analysis of yeast. One of our issue is the difficulty in detecting green fluorophore-tagged proteins, as the fluorophore often disperses widely.

In light of these limitations, we have revised the text.

I found it difficult to follow the rationale for studying class E compartment localizations. These structures accumulate in ESCRT knockdowns in yeast. Do the authors contend that these are stalled intermediates in the formation of a MLVs and is this supported by other literature? Otherwise, it is difficult to see specifically how these inform on MLV biogenesis.

The reviewer brings up important points. In ESCRT mutants, protein trafficking from the endosomal class E compartment to either the vacuole or the Golgi is largely impaired, resulting in the morphological exaggeration of these organelles into stacked cisternae. Class E compartments can revert into MVBs, when cells expressing a *vps4* temperature-sensitive mutant are shifted back from the restrictive to the permissive temperature (Russell et al., 2012). This suggests that Class E compartments represent endosomal structures.

Our data demonstrate that lipid droplets (LDs) form membrane contact sites with Class E compartments. We therefore propose that LDs promote normal endosome biogenesis through the establishment of this membrane contact sites, and may remain bound when endosomes cannot form MVBs. In agreement, we observed that LDs are in close proximity to endosomes containing endocytosed Mup1 (Figure 4G and H). The Class E mutant is thus a first approach to address the role of LDs in endosomal biogenesis.

In addition, conditions where proteins fail to target to the class E compartment based on fluorescence leave unanswered the questions of whether this is a targeting problem or whether the compartment is absent. EM could validate whether the class E compartment still forms in *Vps4*/LD depletion studies.

We have shown this in Figure 6A, B (new Figure 7A, B). As expected, in double mutants (*LDΔvps4Δ*, *vps13Δvps4Δ* and *any1Δvps4Δ*, cells were unable to form Class E compartments and accumulated enlarged compartments next to the vacuole.

Finally, I very much appreciated the discussion on *Vps13* and MVB size/number. The original discovery by Suzuki et al has been difficult to rationalize mechanistically. The authors here make the same observations, and show a similar phenotype in LD knockdown

conditions, supporting their model of a role for Vps13-related LD lipid transport.

Thank you for agreeing with our interpretation.

Minor technical questions:

The image analysis methods are not entirely clear.

1) "% co-localization" is defined as something done in imageJ. Can you provide details on what this metric is (e.g. Pearson's) and the directionality of the co-localization (i.e. X colocalizes with Y may be different from Y colocalizes with X).

Detailed information and quantification methods for the image analysis are now included in the figures and the text.

2) In the case of Mup1, the graphs describe Mup1 dots in the colocalization, presumably because the PM distribution of Mup1 at time 0 would be essentially perfectly colocalized with cortical ER. Please describe how Mup1 dots are identified.

We have added the details in figure legends.

Minor edits:

Line 318 - wrong figure panel is called out

This has been corrected.

Line 459 - I'm not sure you can say "thus scramblase activity is important" without a scramblase specific mutant. Though I agree your data certainly implies a role for scrambling.

We have now identified the specific residues T60 and S64 in Any1, which are located in its scrambling pathway and are essential for scramblase activity *in silico* (Figure 3A) and endosome maturation *in vivo* (Figure 3B-G). These findings support the conclusion that scramblase activity is important for endosome biogenesis.

Reviewer#2

Jieqiong Gao and co-workers show in their manuscript that 'Any1 is a novel phospholipid scramblase involved in endosome biogenesis'. They use a combination of genetic, cell biological, and biochemical experiments and MD simulation to conclude that Any1 functions together with Vps13 to bring lipid droplets close to endosomes, which function together to supply and scramble lipids for proper MVB biogenesis. These findings are interesting and provide novel molecular and cell biological insight into the organization of the endosomal pathway, but additional experiments are required to support these conclusions:

Thank you for the overall positive evaluation of our work.

Major points:

Figure 1: Are the tagged version of Any1 functional?

We conducted growth assays using the *any1* deletion in comparison to cells expressing different tagged versions of Any1 (Figure S1A). We observed a pronounced growth defect in the *any1* Δ mutant on the plates containing 5 mM Zn²⁺, while cells expressing C-terminally tagged Any1 variants grew like wild-type cells (Figure S1A). These findings confirm that the tagged versions of Any1 retain their functionality.

Figure 1: The conclusion that Any1 is a retromer cargo, based on the data is presented, is an overstatement. Either tone down the conclusion, or demonstrate that Any1 is a direct retromer cargo.

We agree with the reviewer and have adjusted our conclusion accordingly.

Figure 1G: Does the Sec7-GFP-Any1-CB complex disrupt Sec7 function, and could that partially explain the more severe growth defect compare to the *any1* mutant. Also is the number of Any1-CB complexes with GFP-Ypt7, Sec7-GFP and GFP-Vps21 comparable? Without this information, the results of this experiment are difficult to interpret.

Our data show that cells expressing Any1-CB did not alter the localization of GFP-tagged Sec7, Vps21, or Ypt7 relative to the vacuole, nor did it affect the number of Vps21 or Ypt7 puncta (Figure S1A, B). However, it did slightly reduce the number of Sec7 puncta (Figure S1C). We agree with the reviewer that we currently lack evidence to demonstrate the functionality of Sec7 in cells expressing Sec7-GFP and Any1-CB. We have therefore revised our conclusion in the text.

Figure 1H: The microscopy of Mup1-GFP need to be complemented either with FACS analysis of Mup1-pHluoring and/or WB analysis. Are other endocytic and biosynthetic MVB cargoes also affect by loss of Any1 function?

We have now complemented the Mup1-GFP sorting assay with an analysis of Mup1-pHluorin (Figure 1J and K), which revealed a similar sorting defect for Mup1 in the *any1* Δ mutant.

To determine whether the absence of Any1 affects other endocytic and biosynthetic MVB cargoes, we monitored Cps1 sorting in both wild-type and *any1* Δ mutant cells. Surprisingly, Cps1 sorting remained unaffected in *any1* Δ cells (now updated as Figure 5E, J), possibly because we monitored the entire pool of Cps1, and predominantly the vacuolar pool, but do not follow the kinetics of its sorting to the vacuole. Additionally, Cps1 is delivered from the Golgi to the endosome, whereas Any1 may function at an earlier stage of endosome formation. For endocytic cargo such as Mup1, we observe a delay in *any1* Δ cells (Figure 1H-K). In agreement, *any1* Δ cells and specific mutants of Any1 that abolish its scramblase activity *in silico* show impaired endosome maturation (Figure 3 and Figure 7). Based on these findings, we conclude that Any1 is important for endosome maturation.

Figure 2: Can Any1 mutants be generated (in silico, in vitro and in vivo) that block lipid scrambling and hence cause a loss of function or a hypomorphic phenotype? These mutants could also be invaluable tools to strength the overall conclusion of the manuscript.

We have identified T60 and S64 as critical residues that are essential for scramblase activity of Any1 *in silico* (Figure 3A) and endosome maturation (Figures 3B-G).

However, due to time constraints of Jieqiong Gao—the first and co-corresponding author of this manuscript, who is moving to Hong Kong to start her own research group in January of 2025, we were unable to further investigate whether these residues are required for scramblase activity *in vitro* or to assess the effects of single mutations *in vivo*. As this analysis requires extensive biochemical analyses and characterization, we decided to follow up on the *in vitro* analysis for a follow up and more extensive study on the mechanism of Any1 in lipid scrambling.

Figure 3B, F: Please indicate the threshold distance for possible MCS.

The threshold distance now has been added to Figures 3B, F.

Figure 3: Based on the current state of the manuscript, the statement that 'LDs form MCS with endosomes' is an overstatement. If anything, there is a spatial correlation of LD with class E compartments. The fact that LD are in close proximity to Mup1 containing endosomes as reported by live cell epifluorescence microscopy means that they could be 200 nm apart from each other. This experiment is insufficient to demonstrate the formation of LD-endosome MCS. Could you block early to late endosome maturation with different mutants and then examine if LD MCS increase.

We agree with the reviewer and have adjusted our conclusion accordingly. As to the other question, this is a very good suggestion, but also tough to address for this revision. Deletions of proteins involved in early endosome biogenesis result in so called Class D vacuoles, which probably correspond to enlarged endosomes, which nevertheless acidify (Conibear and Stevens; Cabrera et al., 2013). Due to the altered vacuole morphology, we are uncertain if the analysis of Class D mutants helps. The Emr group published some temperature sensitive mutants, which may become useful here (Horazdovsky et al., 1996; Burd et al., 2017). As the analysis of LDs and the effect on endocytosis would need extensive work, we decided to postpone this analysis to future studies.

Figure 4 A - H: please provide co-staining with BODIPY and also include the respective any1 single mutant.

We have now included co-staining with BODIPY relative to Cps1 and added the Any1 single mutant in Figure 4A-H (now updated as Figure 5A-I). Our analysis of LD numbers across various mutants showed no significant differences (Figure 5A, B, E-I, and K), suggesting that Any1 and Vps13 are not essential for LD biogenesis.

Figure 5E: in the vps13 mutant, Mup1 is barely internalized, and this could be one reason why less colocalization with LDs is detected. Strictly speaking, this does not mean that Vps13 establishes LD-endosomes MCS.

We agree with the reviewer and have adjusted our conclusion accordingly. However, deletion of *vps4* resulted in Bodipy-positive LDs that were in part close to Class E compartments marked by mCherry-Cps1, in agreement with our TEM analysis (Figure 6I and K, Figure 4D-F). In line with findings on the Mup1 sorting assay (now updated as Figure 6E-H), this MCS was reduced in *vps4Δvps13Δ* cells, whereas Sec63-labeled ER showed significantly more contacts with Class E compartments (now updated as Figure 6I-K). These data support our

model that Vps13 is involved in contact site formation between LDs and endosomes, while additional tether protein(s) might be required to stabilize ER-endosome contact sites.

Figure 6: Based on nice high pressure freezing and TEM experiments, the authors conclude that Any1 and Vps13 are required for appropriate MVB formation. Yet, TEM data does not fully support this conclusion, since measuring MVB diameter on thin sections is difficult to interpret.

We agree that interpreting the data from thin sections alone can be challenging. We therefore aimed to quantify a relatively large number of cells within the limits of electron microscopy (EM), where analyzing large sample sizes can be difficult.

Importantly, our quantification of the *vps13Δ* strain on EM sections aligns well with previously published data (Suzuki et al., 2024), suggesting that our measurements are reliable. Additionally, recent studies have shown that quantification based on thin EM sections can effectively reveal ultrastructural changes in yeast cells (Keuenhof et al., 2022).

Furthermore, the quantification of our light microscopy (LM) data (now updated as Figures 7E and 7F) supports our EM observations. In combination, both LM and EM quantifications support our conclusion that Any1 and Vps13 have critical roles in endosome biogenesis, in agreement with Suzuki et al. (2023).

Also do any1 mutants really have defects in endosomes biogenesis? The MVBs look rather normal (but seem to be less frequent) by electron microscopy. Yet Mup1 preferentially accumulates in endosomes in the any1 mutants. Doesn't that point to a maturation defect along the endosomal compartment?

Indeed, we observed significantly fewer MVBs in the *any1Δ* mutant. Due to the limited number of MVBs, it was challenging to perform a precise quantification to determine whether their morphology matched to that of wild-type cells. However, in *LDΔvps4Δ*, *vps13Δvps4Δ* and *any1Δvps4Δ* cells, Class E compartments failed to form, resulting in the accumulation of enlarged compartments adjacent to the vacuole (now updated as Figure 7B). This observation aligns with our findings on the Cps1 localization (now updated as Figures 5D, F, and H). Taken together, these data indicate that Any1 and Vps13 are involved in proper MVB formation. Therefore, we conclude that Any1 is important for both endosome biogenesis and maturation.

Finally, is there a direct functional relationship between Any1 and Vps13? Do they form a complex on endosomes? Is it clear that any1 functions downstream of Vps13, or could it also be the other way around?

This is a very important point. We performed an ALFA pulldown from Any1-ALFA cells followed by mass spectrometry analysis to identify potential binding partners, but did not observe enrichment of Vps13 (see below). Thus, we currently have no direct evidence that Any1 and Vps13 form a complex. We plan to address this question in a future study by establishing an *in vitro* lipid transfer assay to investigate if Any1 contributes to Vps13's lipid transfer activity.

Figure for reviewers: Label-free proteomics of yeast cells expressing Any1-ALFA compared to untagged control cells. In the plot, the protein abundance ratios of Any1-ALFA over control cells are plotted against the negative log₁₀ of the p value of the two-tailed t test for each protein.

Minor points

Figure 1A - Vps4 instead of 4

Corrected.

Figure 1E - any

Corrected.

Please explain how the colocalization was quantified with image J (Pearson correlation ?)

Detailed information and quantification methods for the image analysis are now included in the figures and the text.

In Figure 1D, does expression of Any1-mNeonGreen- cause an MVB sorting defect? CPS1 appears to accumulate on the limiting vacuolar membrane?

We have now demonstrated that mNeon-tagged Any1 is functional, as shown by the growth test in Figure S1A.

Legend for Figure 1 does not contain sufficient information on image quantification (e.g. 1F)

Detailed information are now included in the figure legends.

The authors conclude: 'Surprisingly, the size of Any1 after SEC was around 158 kDa, while Any1 has a predicted molecular weight of 46.8 kDa (Figure S2 A, B), suggesting that Any1 might form a homo trimer.' Couldn't it be equally well micelle competition?

We agree with the reviewer and have adjusted our conclusion accordingly.

Figure S2D: the green labeling is difficult to see.

This has been changed accordingly for clarity.

Figure 5A -> there is a typo in line 351 (figure 1 is called, instead of Figure 5)

Corrected, thank you.

Reviewer#3

The authors describe the analysis of Any1, a protein that is required for proper MVB formation in yeast. Any1 traffics between Golgi and an endosomal compartment, in part by Retromer sorting, and co-localizes with an early but not with a late-endosome marker. In vitro and in silico studies supported the idea that Any1 functions as scramblase. Electron microscopy imaging suggested that ANY1 deletion affects endosome morphology and MVB numbers. Furthermore, electron microscopy images indicated the presence of contact sites between endosomes and lipid droplets. These contact sites dependent on the presence of Any1 and the lipid transfer protein Vps13. A model is provided that suggests that Any1 is a scramblase at the endosome that is required for proper endosomal maturation by aiding lipid exchange with ER and lipid droplets. Overall, the data is compelling and supports the proposed role of Any1 in the lipid transport to and from the endosome.

Thank you for the overall positive evaluation of our work.

The Mup1 data fit to a simple model in which any1 mutants are delayed in the formation/maturation of MVBs because of lack of sufficient membrane for ILV formation. However, this model does not fit other observations. This simple model would predict an accumulation of immature MVBs and the accumulation of Cps1 in these pre-vacuolar compartments. In contrast, the data show that any1 mutants have less MVB-like structures and when combined with the vps4 mutant, any1 is able to suppress the formation of a class E compartment, which is indicated by the lack of accumulation of Cps1 and membranes in this double mutant. Furthermore, any1 shows less Ypt7 positive late-endosomes. Together, these observations fit better with the idea that loss of Any1 bypasses a checkpoint in endosomal maturation that would normally block immature MVBs from fusing with the vacuole. This checkpoint is the reason why in vps4 endosomal membranes accumulate and form the class E compartment, instead of just fusing with the vacuole. So, loss of Any1 might change lipid composition in a way that promotes fusion?

In summary, the phenotypes of the *any1* mutant, the mutations in the lipid droplets formation and in *Vps13* are complicated, a fact that should be better acknowledged in the discussions. At this point I feel the conclusions are too vague and the complexity of the data cannot be appreciated by researchers that are not intimately familiar with yeast endosomal trafficking.

We thank the reviewer for the valuable comments and constructive criticism.

However, we find it unlikely that the loss of *Any1* promotes fusion of immature endosomes with vacuoles. If this were the case, we would expect the following outcomes, which contradict our observations:

- (1) Faster Mup1 sorting: In contrast, we observed delayed Mup1 sorting in the absence of *Any1* (Figure 1H-K).
- (2) No accumulation of early endosomes: If early endosomes continuously fused with vacuoles, their accumulation would not occur in *Any1* specific mutants that abolish its scramblase activity *in silico* (Figure 3 B-G). However, we clearly observed this phenotype, suggesting that *Any1* is required for endosome maturation.
- (3) No vesicle accumulation in the *any1Δvps4Δ* mutant: If vesicles would directly fuse with vacuoles, vesicle accumulation would be unlikely. However, we observed the accumulated vesicles that failed to form Class E compartments in the *any1Δvps4Δ* mutant (now updated as Figure 7B), indicating that *Any1* is essential for endosome formation.

We thus find our proposed model in better agreement with our data, in which *Any1* function supports both endosome biogenesis and maturation.

Minor point: What is the localization of *Cps1* in *any1* mutant cells?

We thank the reviewer for raising this important point, which was also pointed out by reviewer #2, and the response is listed below:

To determine whether the absence of *Any1* affects other endocytic and biosynthetic MVB cargoes, we monitored *Cps1* sorting in both wild-type and *any1Δ* mutant cells. Surprisingly, *Cps1* sorting remained unaffected in *any1Δ* cells (now updated as Figure 5E, J), possibly because we monitored the entire pool of *Cps1*, and predominantly the vacuolar pool, but do not follow the kinetics of its sorting to the vacuole. Additionally, *Cps1* is delivered from the Golgi to the endosome, whereas *Any1* may function at an earlier stage of endosome formation. For endocytic cargo such as Mup1, we observe a delay in *any1Δ* cells (Figure 1H-K). In agreement, *any1Δ* cells and specific mutants of *Any1* that abolish its scramblase activity *in silico* show impaired endosome maturation (Figure 3 and Figure 7). Based on these findings, we conclude that *Any1* is important for endosome maturation.

References

- Burd et al., 2017 a novel sec18p nsf-dependent-complex-required-for-golgi-to-endosome-transport-in-yeast.
- Cabrera, M., H. Arlt, N. Epp, J. Lachmann, J. Griffith, A. Perz, F. Reggiori, and C. Ungermann. 2013. Functional separation of endosomal fusion factors and the class C

core vacuole/endosome tethering (corvet) complex in endosome biogenesis. *Journal of Biological Chemistry*. 288:5166–5175. doi:10.1074/jbc.M112.431536.

Conibear, E., and T.H. Stevens. Multiple sorting pathways between the late Golgi and the vacuole in yeast.

Horazdovsky, B.F., C.R. Cowles, P. Mustol, M. Holmes, and S.D. Emr. 1996. A novel RING finger protein, Vps8p, functionally interacts with the small GTPase, Vps21p, to facilitate soluble vacuolar protein localization. *Journal of Biological Chemistry*. 271:33607–33615. doi:10.1074/jbc.271.52.33607.

Keuenhof, K.S., L.L. Berglund, S.M. Hill, K.L. Schneider, P.O. Widlund, T. Nyström, and J.L. Höög. 2022. Large organellar changes occur during mild heat shock in yeast. *J Cell Sci*. 135. doi:10.1242/jcs.258325.

Russell, M.R.G., T. Shideler, D.P. Nickerson, M. West, and G. Odorizzi. 2012. Class E compartments form in response to ESCRT dysfunction in yeast due to hyperactivity of the Vps21 Rab GTPase. *J Cell Sci*. 125:5208–5220. doi:10.1242/jcs.111310.

Suzuki, S.W., M. West, Y. Zhang, J.S. Fan, R.T. Roberts, G. Odorizzi, and S.D. Emr. 2024. A role for Vps13-mediated lipid transfer at the ER–endosome contact site in ESCRT-mediated sorting. *Journal of Cell Biology*. 223. doi:10.1083/jcb.202307094.

January 16, 2025

RE: JCB Manuscript #202410013R

Christian Ungermann
Osnabrück University

Dear Prof. Ungermann:

Thank you for submitting your revised manuscript entitled "Any1 is a novel phospholipid scramblase involved in endosome biogenesis". We would be happy to publish your paper in JCB pending final revisions necessary to meet our formatting guidelines (see details below).

A. MANUSCRIPT ORGANIZATION AND FORMATTING:

- 1) Text limits: Character count for Articles is < 40,000, not including spaces. Count includes abstract, introduction, results, discussion, and acknowledgments. Count does not include title page, figure legends, materials and methods, references, tables, or supplemental legends.
- 2) Figures limits: Articles may have up to 10 main text figures.
- 3) Figure formatting: Scale bars must be present on all microscopy images (e.g. Fig 3), including inset magnifications. Molecular weight or nucleic acid size markers must be included on all gel electrophoresis. Aspect ratios of images may not be altered.
- 4) Statistical analysis: Error bars on graphic representations of numerical data must be clearly described in the figure legend. The number of independent data points (n) represented in a graph must be indicated in the legend. Statistical methods should be explained in full in the materials and methods. For figures presenting pooled data the statistical measure should be defined in the figure legends. Please also be sure to indicate the statistical tests used in each of your experiments (either in the figure legend itself or in a separate methods section) as well as the parameters of the test (for example, if you ran a t-test, please indicate if it was one- or two-sided, etc.). Also, if you used parametric tests, please indicate if the data distribution was tested for normality (and if so, how). If not, you must state something to the effect that "Data distribution was assumed to be normal but this was not formally tested."
- 5) Abstract and title: The abstract should be no longer than 160 words and should communicate the significance of the paper for a general audience. The title should be less than 100 characters including spaces. Make the title concise but accessible to a general readership.

* As per JCB policy the term "novel" must be removed from the title
- 6) Materials and methods: Should be comprehensive and not simply reference a previous publication for details on how an experiment was performed. Please provide full descriptions in the text for readers who may not have access to referenced manuscripts.
- 7) All antibodies, cell lines, animals, and tools used in the manuscript should be described in full, including accession numbers for materials available in a public repository such as the Resource Identification Portal. Please be sure to provide the sequences for all of your primers/oligos and RNAi constructs in the materials and methods. You must also indicate in the methods the source, species, and catalog numbers (where appropriate) for all of your antibodies. Please also indicate the acquisition and quantification methods for immunoblotting/western blots.
- 8) Microscope image acquisition: The following information must be provided about the acquisition and processing of images:
 - a. Make and model of microscope
 - b. Type, magnification, and numerical aperture of the objective lenses
 - c. Temperature
 - d. Imaging medium
 - e. Fluorochromes
 - f. Camera make and model
 - g. Acquisition software

h. Any software used for image processing subsequent to data acquisition. Please include details and types of operations involved (e.g., type of deconvolution, 3D reconstitutions, surface or volume rendering, gamma adjustments, etc.).

10) Supplemental materials: There are strict limits on the allowable amount of supplemental data. Articles may have up to 5 supplemental figures. Please also note that tables, like figures, should be provided as individual, editable files. A summary of all supplemental material should appear at the end of the Materials and methods section.

13) ORCID IDs: ORCID IDs are unique identifiers allowing researchers to create a record of their various scholarly contributions in a single place. Please note that ORCID IDs are now *required* for all authors. At resubmission of your final files, please be sure to provide your ORCID ID and those of all co-authors.

Please note that JCB now requires authors to submit Source Data used to generate figures containing gels and Western blots with all revised manuscripts. This Source Data consists of fully uncropped and unprocessed images for each gel/blot displayed in the main and supplemental figures. Since your paper includes cropped gel and/or blot images, please be sure to provide one Source Data file for each figure that contains gels and/or blots along with your revised manuscript files. File names for Source Data figures should be alphanumeric without any spaces or special characters (i.e., SourceDataF#, where F# refers to the associated main figure number or SourceDataFS# for those associated with Supplementary figures). The lanes of the gels/blots should be labeled as they are in the associated figure, the place where cropping was applied should be marked (with a box), and molecular weight/size standards should be labeled wherever possible. Source Data files will be made available to reviewers during evaluation of revised manuscripts and, if your paper is eventually published in JCB, the files will be directly linked to specific figures in the published article.

Journal of Cell Biology now requires a data availability statement for all research article submissions. These statements will be published in the article directly above the Acknowledgments. The statement should address all data underlying the research presented in the manuscript. Please visit the JCB instructions for authors for guidelines and examples of statements at (<https://rupress.org/jcb/pages/editorial-policies#data-availability-statement>).

B. FINAL FILES:

****It is JCB policy that if requested, original data images must be made available to the editors. Failure to provide original images upon request will result in unavoidable delays in publication. Please ensure that you have access to all original data images prior to final submission.****

****The license to publish form must be signed before your manuscript can be sent to production. A link to the electronic license to publish form will be sent to the corresponding author only. Please take a moment to check your funder requirements before choosing the appropriate license.****

Thank you for your attention to these final processing requirements. Please revise and format the manuscript and upload materials within 7 days. If you need an extension for whatever reason, please let us know and we can work with you to determine a suitable revision period.

Thank you for this interesting contribution, we look forward to publishing your paper in Journal of Cell Biology.

Sincerely,

Harald Stenmark, PhD
Monitoring Editor

Andrea L. Marat, PhD
Deputy Editor

Journal of Cell Biology

Reviewer #1 (Comments to the Authors (Required)):

This revised manuscript from Ungermann and colleagues is much improved. Although I continue to believe that EM evidence for the type of contact sites that are inferred here would be ideal, I am satisfied that the additional experiments intended to identify close colocalization at the fluorescence level have strengthened their principle arguments. I also appreciate the new data on Any1 mutants which they speculate are likely to impact scrambling based upon MD studies.

Overall, the role of a scramblase at sites of late endosome membrane dynamics has been anticipated but not shown. I expect this paper will, therefore, garner a wide audience.

Reviewer #2 (Comments to the Authors (Required)):

During the revision of the manuscript, the authors have addressed the most important points, strengthened their conclusions and overall improved the manuscript.

The manuscript now provides an important new perspective on how the fine tuning of the lipid bilayer composition (by combined action of a scrambles (Any1) and a lipid transfer protein (Vps13)) contributes to the biogenesis of multivesicular bodies along the endocytic pathway.